# Can Molecular Foundation Models Know What They Don't Know? A Simple Remedy with Preference Optimization

## Abstract

Molecular foundation models are rapidly advancing scientific discovery, but their unreliability on out-of-distribution (OOD) samples severely limits their application in high-stakes domains such as drug discovery and protein design. A critical failure mode is chemical hallucination, where models make high-confidence yet entirely incorrect predictions for unknown molecules. To address this challenge, we introduce ***Mole**cular **P**reference-**A**ligned **I**nstance **R**anking* (Mole-PAIR), a simple, plug-and-play module that can be flexibly integrated with existing foundation models to improve their reliability on OOD data through cost-effective post-training. Specifically, our method formulates the OOD detection problem as a preference optimization over the estimated OOD affinity between in-distribution (ID) and OOD samples, achieving this goal through a pairwise learning objective. We show that this objective essentially optimizes AUROC, which measures how consistently ID and OOD samples are ranked by the model. Extensive experiments across five real-world molecular datasets demonstrate that our approach significantly improves the OOD detection capabilities of existing molecular foundation models, achieving up to **45.8%**, **43.9%**, and **24.3%** improvements in AUROC under distribution shifts of size, scaffold, and assay, respectively.

## 1 Introduction

Artificial intelligence has enabled major advances in fields such as drug discovery (David et al., 2020) and materials science (Sanchez-Lengeling et al., 2017). Molecular foundation models, pre-trained on large chemical datasets, have shown strong potential to accelerate molecular design by predicting physicochemical and biological properties with improved accuracy (Beaini et al., 2023; Méndez-Lucio et al., 2024; Luo et al., 2023). Despite this promise, a central barrier to their deployment in industrial pipelines is the reliability of their predictions (Jiang et al., 2024; Wang et al., 2024). Without robust confidence estimation, these models may produce misleading outputs.

A key manifestation of this issue is *chemical hallucination*, analogous to hallucination observed in modern large language models (Xu et al., 2025). This problem is particularly severe for out-of-distribution (OOD) molecules (Liang et al., 2017), which deviate significantly from the training distribution. In such cases, models often generate incorrect predictions with high confidence, such as falsely assigning strong bioactivity to an inactive or toxic compound. These failures can have substantial consequences: confidently mispredicting the activity of a novel scaffold may lead to wasted investment in synthesis and testing (Ramsundar et al., 2017), while overlooking an activity cliff—where a small structural change causes a sharp loss of activity (Maggiora, 2006)—can misdirect optimization efforts in drug discovery and design (van Tilborg et al., 2022).

While there are many existing efforts on addressing OOD detection in molecular data or graph data more broadly (Li et al., 2022; Wu et al., 2024; Bao et al., 2024), most of these approaches are tied to specific architectures such as particular graph-based generative models (Liu et al., 2023; Wu et al., 2023) or are directly adapted from images to molecules (Du et al., 2023; Shen et al., 2024; Wang et al., 2025a), which either limits their wide applicability or lacks specific domain knowledge. Furthermore, to the best of our knowledge, the common design of these approaches resorts to pointwise estimation and optimization, where each data sample is allocated with a scalar output indicating its

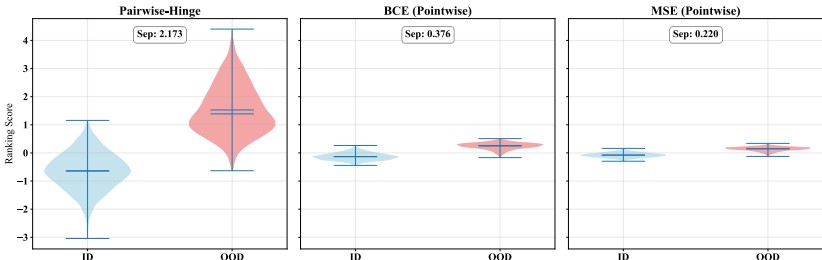

Figure 1: **A case study illustrating the objective-metric misalignment.** The figure plots the estimated OOD affinity scores yielded by the model trained with different objectives for ID and OOD samples on the IC50-Scaffold task. The Pairwise-Hinge loss (Joachims, 2002) produces a globally separated score distribution between ID and OOD, aligning with AUROC, whereas the two pointwise objectives yield heavily overlapping scores due to their per-sample calibration loss. This clearly demonstrates the importance of objective-metric alignment for OOD detection.

affinity to OOD samples and the models are trained with regression-style objectives (Hendrycks & Gimpel, 2016; Liang et al., 2017; Liu et al., 2020; Lee et al., 2018; Sun et al., 2022; Breunig et al., 2000; Li et al., 2025; Zhu et al., 2025). However, this would lead to mismatch with the desired evaluation metric (e.g., AUROC), which seeks the consistent ranking between OOD and ID samples (Figure 1 provides concrete evidence for this issue on real-world data).

To fill this research gap, in this paper, we propose to formulate the OOD detection problem as a preference optimization problem which aims at preserving the correct ranking between any pair of OOD and ID samples. To achieve this goal, we devise a pairwise learning objective, Mole-PAIR, that is agnostic to model architectures and directly optimizes the estimated OOD affinity, without requiring class logits or property labels. Notably, this plug-and-play approach can be seamlessly integrated with arbitrary off-the-shelf molecular foundation models to enhance their OOD detection capability through cost-effective post-training that trains only a lightweight detector. As justification for this design, we prove that this new objective essentially optimizes the AUROC, which measures the consistency of the estimated OOD affinity across any pair of OOD and ID samples. We apply this approach to five public molecular datasets and compare it with recently proposed methods under diverse benchmarking settings including three types of distribution shifts and two recently proposed molecular foundation models (MiniMol (Kläser et al., 2024) and Uni-mol (Zhou et al., 2023; Lu et al., 2024)). The results show that our approach yields consistent improvements across multiple performance metrics, with average gains of 28.3% on AUROC, 28.5% on AUPR, and 25.3% on FPR95 across all datasets.

Our main contributions are summarized below:

- We formulate out-of-distribution detection as a preference optimization problem, where the detector targets consistent ranking of the estimated OOD affinity between in-distribution and out-of-distribution samples. This is achieved by a proposed pairwise learning objective that, as demonstrated by our theoretical analysis, inherently optimizes the AUROC quantifying the ranking consistency across any pair of ID and OOD data.

- On top of this new objective, we frame the proposed approach as a plug-and-play, model-agnostic framework for enhancing the OOD detection capability of molecular foundation models through cost-effective post-training (that does not update the main parameters of pretrained models). This leads to a flexible, lightweight and universal approach for improving the reliability of existing foundation models in molecule-related tasks.

- We conduct comprehensive experiments on multiple challenging molecular OOD benchmarks (DrugOOD (Ji et al., 2022) and GOOD (Gui et al., 2022)) that involve diverse distribution shifts. The results consistently show that Mole-PAIR significantly outperforms recently proposed approaches for OOD detection as measured by AUROC, AUPR, and FPR95, with an average of 28.3% AUROC increase across five datasets and reduction of FPR95 to zero in quite a few cases.

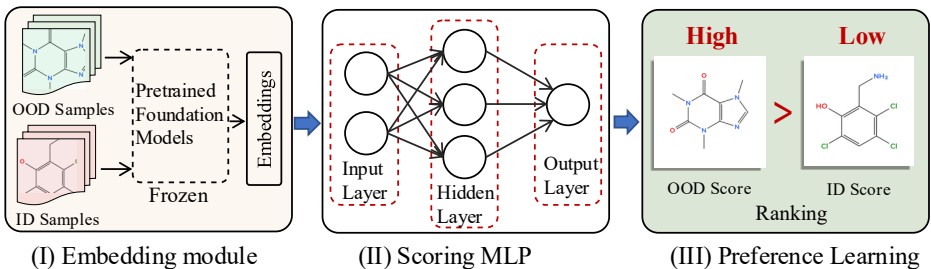

Figure 2: **Overview of the Mole-PAIR framework.**

## 2 RELATED WORK

**General OOD Detection**   OOD detection has attracted wide attention due to the increasing need for building reliable AI. Early approaches such as One-Class SVM (Schölkopf et al., 1999), LOF (Breunig et al., 2000), and Isolation Forest (Liu et al., 2008) operate under Euclidean feature spaces and the i.i.d. assumption. In the deep learning era, widely used generic OOD post-hoc methods include MSP (Hendrycks & Gimpel, 2016) relying on maximum softmax probability, ODIN (Liang et al., 2017) employing temperature scaling and small input perturbations, Mahalanobis distance (Lee et al., 2018) adopting class-conditional Gaussian features, and Deep KNN (Sun et al., 2022), a nonparametric nearest-neighbor detector in the deep feature space. However, these methods are tied to Euclidean embeddings and a softmax classifier head, and they are mostly evaluated on vision benchmarks, which limits their transferability and generalizability to chemistry-constrained molecular graphs and regression tasks.

**OOD Detection for Molecules**   Graph-based models are widely used because many real-world problems involve non-Euclidean relational data, and this has sparked diverse graph OOD detection methods (Li et al., 2022; Guo et al., 2023). These approaches often rely on techniques such as energy-based scoring (Wu et al., 2023), analysis of topological properties (Bao et al., 2024), or synthesis-based OOD generation (Wang et al., 2025a). However, as they are designed for general graphs, these methods often lack domain-specific chemical or molecular knowledge. In contrast, methods that are developed specifically for molecular OOD detection tend to be tightly coupled with specific model architectures and training pipelines, such as reconstruction-similarity with diffusion models for molecules (Shen et al., 2024). This integration makes them less adaptable for use with diverse, pre-trained molecular foundation models, highlighting a need for more flexible solutions.

**AI Reliability**   A key direction for improving AI reliability is to equip pre-trained models with mechanisms to manage predictive uncertainty, especially without costly retraining. Existing post-hoc methods, such as Conformal Prediction and Selective Prediction, focus on managing uncertainty for ID data (Lei et al., 2018; Romano et al., 2020; Lin et al., 2023; Rong et al., 2020; Wang et al., 2025b). Conformal Prediction calibrates a model's scores to produce prediction sets that provably contain the true label with a user-specified frequency, offering distribution-free coverage guarantees for ID samples (Angelopoulos et al., 2022; Laghuvarapu et al., 2023; Arvidsson McShane et al., 2024). Selective Prediction equips a model with a reject option, allowing it to abstain on low-confidence inputs to control the error rate on the predictions it chooses to make (Geifman & El-Yaniv, 2017; 2019; Guo et al., 2017). While they can react to OOD samples, their main goal is to control error rates and manage risk on ID data. In contrast, the fundamental goal of OOD detection is to explicitly identify and flag inputs that originate from a different distribution than the one the model was trained on.

## 3 METHODOLOGY

In this section, we introduce ***Molecular Preference-Aligned Instance Ranking*** (Mole-PAIR), a post-training approach for enhancing the OOD detection capability of molecular foundation models. Our approach casts OOD detection as preference learning and optimizes a pairwise ranking objective that

aligns with optimizing the AUROC between ID and OOD samples. Figure 2 shows the workflow of our proposed model.

## 3.1 OUT-OF-DISTRIBUTION DETECTION FOR MOLECULES

**Problem Formulation.** We consider an input molecule represented by its SMILES string $S \in \mathcal{S}$. We use MiniMol (2D) (Kläser et al., 2024) and Uni-Mol (3D) (Zhou et al., 2023; Lu et al., 2024) to generate molecular embeddings. MiniMol converts the SMILES string into a 2D molecular graph that represents atomic connectivity. This graph, which contains features for atoms and bonds, is then processed by a GNN to produce a final embedding that captures the molecule's topological structure. Uni-mol first generates the molecule's 3D spatial structure from the SMILES string, then uses a transformer-based model to take these 3D coordinates as input to learn a representation that encodes the molecule's geometric shape and spatial properties. For the data setup, we assume access to an ID dataset $D_{\text{in}}^{\text{train}}$ containing molecules with specific bioactive properties of interest. To support pairwise preference learning, we also leverage a set of auxiliary OOD samples, denoted as $D_{\text{out}}^{\text{train}}$. Importantly, we do not assume access to specific OOD distributions encountered at test time. Instead, $D_{\text{out}}^{\text{train}}$ serves as a proxy for the general chemical background, which can be easily sampled from abundant public molecular databases (Irwin et al., 2020; Kim et al., 2025; Mendez et al., 2019), representing 'what is not ID'. This aligns with real-world drug discovery scenarios where specific bioactivity data is scarce, but generic molecular structures are readily available.

Formally, the training set is $D^{train} = D_{in}^{train} \cup D_{out}^{train}$, while at test time, the model is evaluated on $D^{test} = D_{in}^{test} \cup D_{out}^{test}$, where $D_{out}^{test}$ represents novel distribution shifts (e.g., new scaffolds or assays) that are disjoint from the auxiliary training OOD samples as commonly considered in realistic molecular OOD benchmarks.

For the OOD detection task, we aim to design a detector $g$ that decides whether a given molecule $S$ is from in-distribution (ID) or out-of-distribution (OOD):

$$g(S; \tau, E_\phi) = \begin{cases} 1 \ \ (\text{ID}), & \text{if } E_\phi(S) < \tau, \\ 0 \ \ (\text{OOD}), & \text{if } E_\phi(S) \geq \tau, \end{cases} \tag{1}$$

where $E_\phi(\cdot)$ is a learnable detector and $\tau$ is a decision threshold. The score from $E_\phi(S)$ indicates the OOD affinity of the sample $S$ estimated by the model. The ideal detector should satisfy $E_\phi(S_{\text{in}}) < E_\phi(S_{\text{out}})$ for all $S_{\text{in}} \sim D_{\text{in}}$ and $S_{\text{out}} \sim D_{\text{out}}$.

**OOD Detector for Foundation Models.** Without loss of generality, we consider a molecular foundation model $f_{\text{Encoder}}$ that maps any input molecule $S$ to an embedding vector in latent space:

$$h = f_{\text{Encoder}}(S) \in \mathbb{R}^p. \tag{2}$$

We compose the scoring function $E_\phi$ from a frozen, pre-trained $f_{\text{Encoder}}$ and a trainable head $g_{\text{Head}}$:

$$E_\phi(S) = g_{\text{Head}}\big(f_{\text{Encoder}}(S); \phi\big), \tag{3}$$

where the trainable parameters $\phi$ consist only of those in $g_{\text{Head}}$ while the parameters of $f_{\text{Encoder}}$ are frozen. This plug-in design focuses post-training on the OOD scoring function (also known as the detector) and is compatible with any off-the-shelf molecular foundation models.

## 3.2 PREFERENCE OPTIMIZATION LEARNING OBJECTIVE

Achieving the goal of $E_\phi(S_{\text{in}}) < E_\phi(S_{\text{out}})$ is non-trivial since such a target involves the ranking between any pair of ID and OOD samples. A quantitative metric for measuring how close the ranking yielded by the model's estimation is to the ideal case is the Area Under the Receiver Operating Characteristic Curve (AUROC) that equals the probability that for any pair of ID and OOD samples how likely the ID sample is ranked before the OOD one:

$$\text{AUROC} = \Pr\big(E_\phi(S_{\text{in}}) < E_\phi(S_{\text{out}})\big), \quad S_{\text{in}} \sim D_{\text{in}}^{\text{test}}, \ S_{\text{out}} \sim D_{\text{out}}^{\text{test}}. \tag{4}$$

The ideal learning objective for OOD detection should maximize the AUROC produced by the model's estimated OOD affinity.

**Preference Optimization for OOD Detection.** Directly optimizing AUROC is intractable due to the non-differentiability. In this work, we resort to a pairwise learning objective inspired by direct preference optimization (DPO) (Rafailov et al., 2024) that has shown success in post-training modern large language models (LLMs) (Wang et al., 2023; Tunstall et al., 2023; Hong et al., 2024; Meng et al., 2024). The main idea of DPO is to learn directly from pairwise preferences with a simple logistic likelihood, without training a separate reward model or using on-policy reinforcement learning for LLM post-training. We extend this principle for post-training molecular foundation models for OOD detection, where, in particular, we replace the LLM policy score with the estimated OOD affinity $E_\phi(S)$ produced by the detector head on top of a frozen encoder, and use only the pairwise ranking labels for optimization. Specifically, following the Bradley–Terry model (Bradley & Terry, 1952), we introduce a pairwise learning objective that imposes the target that for any pair of ID and OOD samples, the ID sample is preferred:

$$P(S_{\text{in}} \succ S_{\text{out}}) = \sigma\big(\beta \cdot [E_\phi(S_{\text{out}}) - E_\phi(S_{\text{in}})]\big), \tag{5}$$

where $\sigma(u) = (1 + e^{-u})^{-1}$ is the logistic sigmoid that converts the detector margin $\Delta E_\phi = E_\phi(S_{\text{out}}) - E_\phi(S_{\text{in}})$ into a probability in $[0, 1]$. The larger the margin, the closer this probability is to 1, indicating stronger confidence that the ID sample should be ranked before the OOD sample. The temperature parameter $\beta > 0$ adjusts how sensitive the probability is to the margin: a larger $\beta$ makes the model react more strongly to small differences, whereas a smaller $\beta$ yields more gradual training signals. Maximizing the log-likelihood over pairs yields the **Mole-PAIR loss**:

$$\mathcal{L}_{\text{Mole-PAIR}}(\phi) = - \mathbb{E}_{S_{\text{in}} \sim D_{\text{in}}, \, S_{\text{out}} \sim D_{\text{out}}} \left[ \log \sigma\big(\beta \cdot [E_\phi(S_{\text{out}}) - E_\phi(S_{\text{in}})]\big) \right]. \tag{6}$$

In practice we construct balanced mini-batches of ID and OOD samples and compute margins within the batch to estimate the expectation efficiently. This focuses learning on relative ranking rather than absolute scores and is consistent with optimizing AUROC (see more justification in Sec. 4).

**Regularization.** Since AUROC depends only on the ordering of scores and the loss function Eq. (6) depends only on the margin, adding a constant offset to the detector output does not change the overall trend. Thus, the training objective is invariant to global translations $E_\phi(S) \mapsto E_\phi(S) + c$, which can cause scale drift and numerical instability. To fix this gauge and stabilize the scale without affecting the optimal ranking, we add a small $\ell_2$ penalty on the detector output:

$$\mathcal{L}_{\text{total}}(\phi) = \mathcal{L}_{\text{Mole-PAIR}}(\phi) + \lambda\big(\mathbb{E}[E_\phi(S_{\text{in}})^2] + \mathbb{E}[E_\phi(S_{\text{out}})^2]\big), \qquad \lambda > 0 \text{ small.} \tag{7}$$

This penalty anchors the mean OOD affinity near zero and discourages unnecessarily large magnitudes, preventing saturation of the loss function when $\beta \Delta E_\phi$ becomes too large. At test time we use $E_\phi(S)$ as the OOD affinity (higher means more OOD-like) and evaluate the model with multiple threshold-free metrics including AUROC on $(D_{\text{in}}^{\text{test}}, D_{\text{out}}^{\text{test}})$.

## 4 Theoretical Discussions

In this section, we provide the theoretical justification for Mole-PAIR and show that our preference-based framework is formally principled. To begin with, we clarify the mismatch between traditional point-wise estimation and the desired evaluation metric in OOD detection. What OOD detection ultimately evaluates is a ranking: the AUROC equals $\Pr\big(E_\phi(S_{\text{in}}) < E_\phi(S_{\text{out}})\big)$ (Fawcett, 2006; Cortes & Mohri, 2003), i.e., how often ID scores rank below OOD scores. Pointwise objectives such as using MSE or BCE to regress a single scalar score assume absolute calibration, which is fragile under distribution shifts and orthogonal to AUROC's relative nature. Mole-PAIR instead trains on pairwise ID–OOD comparisons via the logistic loss in Eq. 6 and the total objective in Eq. 7. During training, the encoder used for molecular embeddings stays frozen and we only learn a small ranking score head $E_\phi(\cdot)$, which makes post-training lightweight and property-label free. Notation, assumptions, and the full objective are summarized in Appendix B.1.

### 4.1 Adaptive Learning by Prioritizing Hard Pairs

We now unpack what this loss optimizes in practice. For any ID–OOD pair $(S_{\text{in}}, S_{\text{out}})$, we define:

$$\Delta E_\phi = E_\phi(S_{\text{out}}) - E_\phi(S_{\text{in}}). \tag{8}$$

This is the margin between ID and OOD ranking scores. The logistic term $\log\big(1 + \exp(-\beta\,\Delta E_\phi)\big)$ is small when the ordering is correct with an appropriate gap, large when the pair is misranked, and steepest near the decision boundary. Thus, the loss translates AUROC's ranking objective into gradient updates that concentrate on pairs misordered or close to being misordered. To formalize this intuition, we examine the gradient and find it naturally emphasizes misranked or borderline pairs. Additional gradient and curvature derivations are provided in Appendix B.2.

*Proposition* 4.1 (Hard-pair emphasis). Let $d_\phi = \nabla_\phi E_\phi(S_{\text{out}}) - \nabla_\phi E_\phi(S_{\text{in}})$. A gradient step of size $\eta > 0$ on Eq. 6 changes the margin by

$$\delta(\Delta E_\phi) \;\approx\; \eta\,\beta\,\sigma(-\beta\,\Delta E_\phi)\,\|d_\phi\|_2^2 \;\geq\; 0, \tag{9}$$

where $\sigma(u) = (1 + e^{-u})^{-1}$. The weight $\sigma(-\beta\,\Delta E_\phi)$ decreases with $\Delta E_\phi$, which means it is largest for misranked or borderline pairs and smallest for already separated pairs. The detailed proof is provided in Appendix B.3 and additional experiments are provided in Appendix D.

Proposition 4.1 reveals the intuition behind Mole-PAIR. During training, the model naturally focuses more on the pairs that degrade AUROC the most—those that are misranked or near the boundary. These pairs receive stronger gradient updates, enlarging their margins first and further separating ID from OOD samples. Thus, the dynamics are self-correcting: once a pair is confidently ranked, it no longer consumes optimization effort, and the model shifts attention to the remaining hard cases. This aligns with the nature of AUROC, which is only affected by misranked or borderline pairs, and ensures that training resources are allocated to the most critical regions of the score distribution. For further practical details on the gradient behavior, see Appendix B.7.

## 4.2 Convergence to the Optimal Ranking

The analysis above explains how our training objective dynamically prioritizes difficult pairs. We now demonstrate that this process converges to a globally optimal solution. We will prove that, given sufficient data and model capacity, the learned scorer achieves the Bayes-optimal ranking.

We define an ID-preference score:

$$f(S) \triangleq -E_\phi(S), \tag{10}$$

so that larger $f$ means "more ID-like" (equivalently, lower ranking score). For any two samples $S, S'$, the pairwise score margin is:

$$z(S, S') \;\triangleq\; f(S) - f(S') \;=\; E_\phi(S') - E_\phi(S). \tag{11}$$

Thus $z > 0$ means that $S$ is ranked ahead of $S'$ as ID (i.e., $E_\phi(S) < E_\phi(S')$).

*Lemma* 4.2 (Local Pairwise Optimality). Let $\eta(S) = \Pr(\text{ID} \mid S)$ be the true posterior probability that a sample $S$ is in-distribution. For any two samples $S, S'$, consider the conditional pairwise risk for $z = f(S) - f(S')$:

$$r_\beta(z; S, S') = \eta(S)\big(1 - \eta(S')\big) \log\big(1 + e^{-\beta z}\big) + \eta(S')\big(1 - \eta(S)\big) \log\big(1 + e^{\beta z}\big). \tag{12}$$

Then $r_\beta(z; S, S')$ is strictly convex in $z$ and is minimized at:

$$z^\star = \beta^{-1} \log \frac{\eta(S)\,[1 - \eta(S')]}{\eta(S')\,[1 - \eta(S)]}, \tag{13}$$

whose sign matches that of $\eta(S) - \eta(S')$. The detailed proof is provided in Appendix B.4.

Lemma 4.2 shows that for any pair $(S, S')$, the optimal score margin $z^\star$ aligns perfectly with the true posterior probabilities. If $S$ is more likely to be ID than $S'$ (i.e., $\eta(S) > \eta(S')$), the optimal margin $z^\star$ is positive, correctly ranking $S$ higher. If both are equally likely, the optimal margin is zero, ensuring the model does not impose an artificial preference on an ambiguous pair. Furthermore, the magnitude of this optimal gap, $|z^\star|$, scales with the confidence in the ordering (i.e., the distance between posteriors), while the temperature $\beta$ simply rescales this gap without altering the ranking.

*Proposition* 4.3 (Global Convergence to the Bayes-Optimal Ranking). Define the pairwise risk of a scorer $f$ by:

$$\mathcal{R}_\beta(f) \;=\; \mathbb{E}_{(S_{\text{in}}, S_{\text{out}})}\Big[ \log\big(1 + e^{-\beta\,[\,f(S_{\text{in}}) - f(S_{\text{out}})\,]}\big)\Big], \tag{14}$$

where the expectation is over independent draws $S_{\text{in}} \sim D_{\text{in}}$ and $S_{\text{out}} \sim D_{\text{out}}$. For any sufficiently expressive function class, every global minimizer $f^\star$ of $\mathcal{R}_\beta$ induces the same ordering as $\eta(\cdot)$ for almost all pairs, and hence achieves the Bayes-optimal AUROC (Detailed proof in Appendix B.5).

Lemma 4.2 establishes the optimal behavior for a single pair of samples, while Proposition 4.3 generalizes this local result to the entire data distribution. The proposition asserts that optimizing the global risk—an expectation over all ID-OOD pairs—drives the scorer towards the Bayes-optimal ranking. Since the global objective is an aggregate of these pairwise terms, any scorer that systematically misranks a set of pairs can be improved by adjusting its scores towards the local optima defined in Lemma 4.2. Therefore, with sufficient model capacity and data, minimizing our pairwise objective recovers the true ranking induced by $\eta(\cdot)$, which by definition maximizes the AUROC. For the $\beta \to \infty$ asymptotics and the link to 0–1 ranking, see Appendix B.6.

## 5 EXPERIMENTS

Mole-PAIR can be applied to OOD detection across diverse tasks as a single model. In this section, we present empirical evidence to validate the effectiveness of the Mole-PAIR framework. The goal of our experiments is to demonstrate the practical efficacy of our approach in enhancing the OOD detection capabilities of existing molecular foundation models, rather than outperforming state-of-the-art methods specifically tailored for molecular OOD detection.

**Datasets.** With increasing attention on molecular OOD detection, two key benchmarks have been proposed: DrugOOD (Ji et al., 2022) and GOOD (Gui et al., 2022). DrugOOD is a systematic OOD dataset curator and benchmark for drug discovery, offering large-scale, realistic, and diverse datasets. To cover a variety of shifts that naturally occur in molecules, we cautiously selected three properties to divide the ID and OOD data: assay, molecular size, and molecular scaffold, respectively. GOOD is another systematic OOD benchmark that provides carefully designed data environments for distribution shifts. From this benchmark, we mainly consider covariate shift in our experiments.

**Evaluation Metrics.** In the experiments, we mainly report the following metrics: (1) the area under the receiver operating characteristic curve (AUROC), (2) the area under the precision-recall curve (AUPR), and (3) the false positive rate FPR95 of OOD samples when the true positive rate of ID samples is $95\%$. More baseline details can be found in Appendix C.2.

**Training Details.** We use pretrained MiniMol (Kläser et al., 2024) (2D graph features) and Uni-Mol (Zhou et al., 2023; Lu et al., 2024) (3D conformational features) as frozen encoders, each yielding a fixed 512-dimensional representation per molecule. Our Mole-PAIR module attaches a lightweight MLP scoring head with structure $512 \to 256 \to 128 \to 1$ and dropout 0.1; this head is the only trainable part. Mole-PAIR is trained without class labels, using only ID/OOD pairing signals. We optimize with AdamW (learning rate $1 \times 10^{-4}$, weight decay $1 \times 10^{-5}$), apply a StepLR scheduler (step size 10, $\gamma = 0.9$), and clip gradients at norm 1.0. The temperature is $\beta = 0.1$. Training runs for up to 500 epochs with batch size 512 for MiniMol and 256 for Uni-mol; inference uses the corresponding encoder-specific batch sizes.

**Baselines.** For supervised baselines we attach a classifier head to the same frozen 512-dimensional features. The head is an MLP with three layers and the hidden dimension is set to 64. The loss is CrossEntropyLoss for single-task classification, BCEWithLogitsLoss for multi-task classification, and MSELoss for regression when present prior to binarization. For datasets with continuous targets such as GOOD-ZINC, we binarize by a median split (label 1 if the value is $\geq$ median, else 0) and train the classifier accordingly. Optimization uses Adam with learning rate 0.01 and weight decay $5 \times 10^{-4}$ for 500 epochs, with early stopping (patience $= 30$) and best-validation checkpointing. To ensure fairness, when baselines are evaluated together we reuse the same trained checkpoint for scoring; when trained separately, each baseline uses the same architecture, hyperparameters, and splits. Based on this backbone, OOD scores are computed using MSP (Hendrycks & Gimpel, 2016), ODIN (Liang et al., 2017), Energy (Liu et al., 2020), Mahalanobis (Lee et al., 2018), LOF (Breunig et al., 2000), and KNN (Sun et al., 2022) as defined in the baseline section in Appendix C.2.

### 5.1 OVERALL PERFORMANCE

Our experimental results, summarized for AUROC in Table 1 and FPR95 in Table 2, reveal a fundamental limitation of standard baselines for molecular out-of-distribution detection. For comprehensive results including AUPR, see Table 11 in Appendix D. Confidence-based approaches such as

Table 1: Out-of-distribution detection results measured by **AUROC** (↑). The best results of all methods are indicated in boldface, and the second best results are underlined.

| | EC50 | | | IC50 | | | HIV | | PCBA | | ZINC | |
|---|---|---|---|---|---|---|---|---|---|---|---|---|
| | Scaffold | Size | Assay | Scaffold | Size | Assay | Scaffold | Size | Scaffold | Size | Scaffold | Size |
| **MiniMol** | | | | | | | | | | | | |
| MSP | 0.677 | 0.449 | 0.420 | 0.574 | 0.515 | 0.600 | 0.408 | 0.194 | 0.632 | 0.697 | 0.359 | 0.398 |
| ODIN | 0.633 | 0.450 | 0.437 | 0.575 | 0.516 | 0.614 | 0.390 | 0.336 | 0.623 | 0.691 | 0.360 | 0.398 |
| Energy | 0.685 | 0.527 | 0.455 | 0.562 | 0.573 | 0.569 | 0.388 | 0.167 | 0.642 | 0.735 | 0.359 | 0.398 |
| Mahalanobis | 0.660 | 0.831 | 0.572 | 0.620 | 0.751 | 0.516 | 0.503 | 0.918 | 0.564 | 0.728 | 0.638 | 0.593 |
| LOF | 0.665 | 0.838 | 0.537 | 0.625 | 0.745 | 0.564 | 0.508 | 0.889 | 0.564 | 0.687 | 0.544 | 0.688 |
| KNN | 0.671 | 0.855 | 0.569 | 0.655 | 0.765 | 0.502 | 0.562 | 0.921 | 0.459 | 0.527 | 0.636 | 0.676 |
| Mole-PAIR | **0.970** | **1.000** | **0.711** | **0.983** | **0.999** | **0.660** | **0.777** | **1.000** | **0.924** | **1.000** | 0.614 | **1.000** |
| *Improvement* | ↑ 41.6% | ↑ 17.0% | ↑ 24.3% | ↑ 50.1% | ↑ 30.6% | ↑ 7.5% | ↑ 38.3% | ↑ 8.6% | ↑ 43.9% | ↑ 36.1% | ↓ −3.76% | ↑ 45.4% |
| **Uni-mol** | | | | | | | | | | | | |
| MSP | 0.694 | 0.722 | 0.547 | 0.585 | 0.637 | 0.549 | 0.465 | 0.218 | 0.476 | 0.350 | 0.352 | 0.369 |
| ODIN | 0.621 | 0.656 | 0.558 | 0.546 | 0.591 | 0.538 | 0.451 | 0.278 | 0.445 | 0.353 | 0.352 | 0.369 |
| Energy | 0.691 | 0.652 | 0.547 | 0.542 | 0.598 | 0.551 | 0.476 | 0.243 | 0.476 | 0.350 | 0.352 | 0.369 |
| Mahalanobis | 0.724 | 0.784 | 0.591 | 0.699 | 0.723 | 0.562 | 0.567 | 0.826 | 0.568 | 0.690 | 0.643 | 0.643 |
| LOF | 0.748 | 0.855 | 0.558 | 0.717 | 0.759 | 0.547 | 0.594 | 0.860 | 0.538 | 0.607 | 0.577 | 0.621 |
| KNN | 0.704 | 0.747 | 0.599 | 0.682 | 0.660 | 0.575 | 0.580 | 0.826 | 0.553 | 0.668 | 0.645 | 0.686 |
| Mole-PAIR | **0.965** | **1.000** | **0.650** | **0.977** | **1.000** | **0.640** | **0.728** | **1.000** | **0.875** | **1.000** | 0.549 | **1.000** |
| *Improvement* | ↑ 29.0% | ↑ 17.0% | ↑ 8.5% | ↑ 36.3% | ↑ 31.8% | ↑ 11.3% | ↑ 22.6% | ↑ 16.3% | ↑ 54.1% | ↑ 44.9% | ↓ −14.9% | ↑ 45.8% |

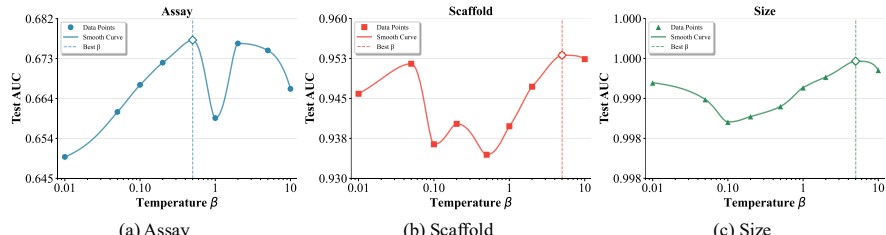

(a) Assay        (b) Scaffold        (c) Size

Figure 3: **Test AUROC sensitivity to the temperature $\beta$ with $\lambda = 0.01$.** Different distribution shifts show distinct sensitivities: **Assay** prefers a medium $\beta$, **Scaffold** favors a larger $\beta$, while **Size** is largely insensitive to the choice of $\beta$.

MSP and ODIN consistently underperform under realistic distribution shifts. For example, on the MiniMol–EC50–Scaffold split, MSP yields an AUROC of 0.677 with an FPR95 of 0.725, and performance further deteriorates on the HIV–Scaffold split with an AUROC of 0.408 and an FPR95 of 0.953, where the model assigns high confidence to almost all novel OOD molecules. More advanced density- and distance-based methods offer occasional gains but remain unstable and unreliable.

In contrast, Mole-PAIR consistently surpasses all baselines across datasets, distribution shifts, and model backbones. On MiniMol–EC50–Scaffold, Mole-PAIR raises the AUROC from 0.677 to 0.970 and reduces the FPR95 from 0.725 to 0.178. On size-based splits, it often achieves near-perfect results with AUROC close to 1.000 and FPR95 near 0.000. Even on the most challenging setting such as HIV–Scaffold, Mole-PAIR delivers substantial improvements. These gains hold for both the lightweight MiniMol and the powerful Uni-mol encoder, underscoring its broad applicability.

Overall, these findings demonstrate that Mole-PAIR directly addresses the central challenge of molecular OOD detection, namely chemical hallucination. By optimizing the relative ranking between ID and OOD samples rather than relying on absolute confidence scores, it effectively reduces false positives, for instance lowering FPR95 from above 0.90 to below 0.20 on assay splits. As a lightweight, label-free, and backbone-agnostic framework, Mole-PAIR transforms molecular foundation models into reliable and robust tools for high-stakes applications such as drug discovery.

## 5.2 ABLATION STUDIES

**The effect of temperature parameter $\beta$.** We conduct experiments over various $\beta$ in Eq. 6 while keeping the $\ell_2$ regularization $\lambda$ fixed at 0.01, and the experimental results are shown in Figure 3. Specifically, we list the test AUC performance with different values of $\beta$ ranging from 0.01 to 10.0 on the EC50 dataset, under the Scaffold, Size, and Assay distribution shifts. From the figure, we find

Table 2: Comparison of OOD detection performance in terms of FPR95 (↓). Lower values indicate better detection performance. Results are reported across different datasets and distribution shifts. Best results are in **bold**, and second-best are underlined.

| | EC50 | | | IC50 | | | HIV | | PCBA | | ZINC | |
|---|---|---|---|---|---|---|---|---|---|---|---|---|
| | Scaffold | Size | Assay | Scaffold | Size | Assay | Scaffold | Size | Scaffold | Size | Scaffold | Size |
| **MiniMol** | | | | | | | | | | | | |
| MSP | 0.725 | 0.833 | 0.972 | 0.759 | 0.743 | 0.903 | 0.953 | 0.983 | 0.833 | 0.698 | 0.985 | 0.946 |
| ODIN | 0.734 | 0.832 | 0.950 | 0.563 | 0.741 | 0.900 | 0.962 | 0.819 | 0.856 | 0.763 | 0.985 | 0.947 |
| Energy | 0.716 | 0.726 | 0.973 | 0.781 | 0.662 | 0.905 | 0.955 | 0.985 | 0.832 | 0.697 | 0.985 | 0.946 |
| Mahalanobis | 0.898 | 0.633 | 0.929 | 0.909 | 0.786 | 0.946 | 0.947 | 0.315 | 0.922 | 0.752 | **0.905** | 0.874 |
| LOF | 0.893 | 0.667 | 0.933 | 0.909 | 0.803 | 0.934 | 0.943 | 0.458 | 0.934 | 0.804 | 0.933 | 0.797 |
| KNN | 0.870 | **0.550** | 0.930 | 0.898 | 0.782 | 0.942 | 0.906 | 0.280 | 0.960 | 0.902 | 0.906 | 0.801 |
| Mole-PAIR | **0.178** | **0.000** | **0.823** | **0.084** | **0.004** | **0.861** | **0.624** | **0.001** | **0.348** | **0.001** | 0.925 | **0.000** |
| *Improvement* | ↑ 75.4% | ↑ 100.0% | ↑ 11.4% | ↑ 85.1% | ↑ 99.4% | ↑ 4.3% | ↑ 31.1% | ↑ 99.7% | ↑ 58.2% | ↑ 99.9% | ↓ −2.21% | ↑ 100.0% |
| **Uni-mol** | | | | | | | | | | | | |
| MSP | 0.848 | 0.733 | 0.957 | 0.926 | 0.861 | 0.945 | 0.954 | 0.985 | 0.954 | 0.925 | 0.992 | 0.996 |
| ODIN | 0.816 | 0.798 | 0.913 | 0.945 | 0.911 | 0.943 | 0.947 | 0.977 | 0.962 | 0.925 | 0.993 | 0.997 |
| Energy | 0.849 | 0.740 | 0.948 | 0.935 | 0.872 | 0.949 | 0.952 | 0.982 | 0.954 | 0.925 | 0.993 | 0.996 |
| Mahalanobis | 0.831 | 0.605 | 0.921 | 0.818 | 0.718 | 0.936 | 0.920 | 0.514 | 0.836 | 0.855 | 0.965 | 0.938 |
| LOF | 0.844 | 0.554 | 0.935 | 0.832 | 0.746 | 0.942 | 0.894 | 0.567 | 0.948 | 0.926 | **0.945** | 0.919 |
| KNN | 0.855 | 0.707 | 0.914 | 0.833 | 0.806 | 0.929 | 0.885 | 0.563 | 0.941 | 0.895 | 0.961 | 0.899 |
| Mole-PAIR | **0.178** | **0.000** | **0.869** | **0.139** | **0.000** | **0.890** | **0.736** | **0.000** | **0.515** | **0.000** | 0.949 | **0.000** |
| *Improvement* | ↑ 78.2% | ↑ 100% | ↑ 4.8% | ↑ 83.1% | ↑ 100% | ↑ 4.2% | ↑ 16.8% | ↑ 100% | ↑ 38.4% | ↑ 100% | ↓ −0.42% | ↑ 100% |

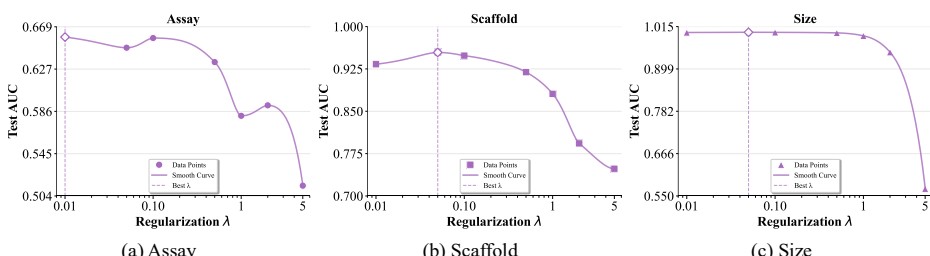

(a) Assay      (b) Scaffold      (c) Size

Figure 4: **Test AUROC sensitivity to the $\ell_2$ regularization $\lambda$ with $\beta = 0.1$.** Performance varies with regularization strength: **Assay** performs best with weak regularization, **Scaffold** benefits from a modest amount of regularization, while **Size** is robust until the regularization becomes too strong.

that different distribution shifts exhibit distinct sensitivities to the choice of $\beta$. The performance on the Scaffold split is highly sensitive to $\beta$, showing a non-monotonic trend where the optimal performance (AUC ≈ 0.953) is achieved at a relatively large value ($\beta$=5.0). In contrast, the performance on the Size split is robust and largely insensitive to $\beta$, achieving near-perfect AUC scores across the entire range of values. The performance on the Assay split also shows moderate sensitivity, peaking at $\beta$=0.5. This not only demonstrates the effectiveness of our Mole-PAIR framework but also reveals that complex shifts, like scaffold difference, are harder to learn and benefit from a stronger preference optimization signal, whereas simpler shifts based on molecular size are easily separable regardless of the signal strength.

**The influence of regularization parameter $\lambda$.** We further analyze the effect of the regularization coefficient $\lambda$ while keeping the DPO temperature $\beta$ fixed at 0.1. The results across the three distribution shifts on the EC50 dataset are shown in Figure 4. The datasets demonstrate varied sensitivities to $\lambda$. For the Assay split, performance is optimal with a small regularization ($\lambda \leq 0.1$) and degrades steadily as the penalty increases. The Scaffold split exhibits a clearer trend, with performance peaking at an optimal value of $\lambda$=0.05 before declining sharply, indicating that a modest amount of regularization is beneficial for this complex task. Conversely, the Size split is robust to a wide range of regularization strengths, maintaining near-perfect performance for $\lambda$ up to 1.0, after which it drops precipitously. Overall, these findings highlight the role of $\lambda$ as a critical hyperparameter. While an optimal value can improve generalization on challenging shifts like Scaffold, an excessively large regularization coefficient is consistently detrimental to performance across all tasks, likely due to over-constraining the ranking function and preventing effective separation.

### 5.3 GENERALIZATION TO UNSEEN OOD DOMAINS

A key concern in OOD detection is whether the method relies on seeing specific OOD patterns during training. To verify that Mole-PAIR learns to define the ID boundary rather than overfitting to the training OOD samples, we conducted rigorous cross-domain generalization experiments. We trained the model using one type of distribution shift (e.g., Scaffold split) as the auxiliary OOD source and evaluated it on a completely different, unseen shift (e.g., Size split).

As detailed in Appendix D.3, Mole-PAIR maintains superior performance even when the testing OOD distribution is structurally distinct from the training 'auxiliary' OOD data. This empirical evidence confirms that our use of auxiliary OOD data successfully regularizes the model to reject samples outside the ID manifold, proving its robustness in realistic scenarios where future OOD types are unknown.

### 5.4 ANALYSIS OF TRAINING DYNAMICS

**Self-paced learning behavior.** Across all three shifts, the dynamics match our theoretical analysis that the proposed Mole-PAIR prioritizes hard and borderline pairs. Concretely, both the misranked proportion $\Pr(\Delta E_\phi < 0)$ and the boundary mass $\Pr(|\Delta E_\phi| < \varepsilon)$ drop rapidly during the first few epochs, while the mean margin $\mathbb{E}[\Delta E_\phi]$ increases steadily throughout training. This behavior is predicted by Eq. 9. Gradient updates are weighted by $w_\beta(\Delta) = \beta\,\sigma(-\beta\Delta)$, which is largest for misranked or borderline pairs and vanishes for already well-separated ones; hence the optimizer first fixes the pairs that most degrade AUROC and then spends diminishing effort on the rest.

**Shift-specific dynamics.** The three splits exhibit distinct rates of separation, consistent with our ablations: (i) **Size** corrects fastest: both error and boundary mass collapse early, and $\mathbb{E}[\Delta E_\phi]$ becomes large, indicating that size-based OOD is geometrically easy once the head has been trained. (ii) **Scaffold** improves steadily but more slowly, requiring more epochs to push borderline pairs away from the decision boundary—consistent with our $\beta$-sensitivity study, where Scaffold prefers a larger temperature (a sharper preference signal). (iii) **Assay** is the hardest: the misranked proportion diminishes but plateaus at a higher level; the boundary mass decreases more gradually; and the margin grows but remains comparatively small—again aligned with our observation that Assay favors a moderate $\beta$ and weak regularization $\lambda$.

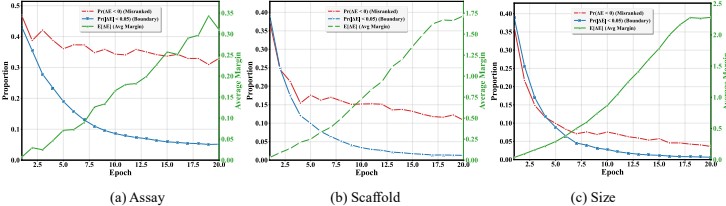

(a) Assay       (b) Scaffold       (c) Size

Figure 5: **Training dynamics of Mole-PAIR across three distribution shifts.** Each panel corresponds to one shift—(a) Assay, (b) Scaffold, (c) Size—and plots three metrics over 20 epochs: the misranked-pair proportion $\Pr(\Delta E_\phi < 0)$ (left $y$-axis), the boundary mass $\Pr(|\Delta E_\phi| < \varepsilon)$ with $\varepsilon = 0.05$ (left $y$-axis), and the average margin $\mathbb{E}[\Delta E_\phi]$ (right $y$-axis), where $\Delta E_\phi = E_\phi(S_{\text{out}}) - E_\phi(S_{\text{in}})$. The rapid decrease of the first two curves and the steady increase of the margin illustrate that hard or borderline pairs are corrected first.

## 6 CONCLUSION

In this paper, we propose Mole-PAIR, a lightweight, plug-and-play, and model-agnostic framework that can be flexibly integrated with existing foundation models to endow them with OOD detection capabilities. Unlike conventional approaches which resort to point-wise estimation and optimization, we innovatively reframe the OOD detection problem as a preference optimization problem by carefully devising a pairwise loss. We theoretically justify that this pairwise learning objective aligns with the AUROC metric, which measures how consistently the model ranks ID samples higher than OOD samples. Extensive experiments on five real-world datasets under three different kinds of distribution shifts demonstrate the effectiveness and superiority of our model.

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

## A  LARGE LANGUAGE MODEL USAGE STATEMENT

We acknowledge the use of a large language model (LLM) as an auxiliary tool in the preparation of this manuscript. The role of the LLM was strictly confined to language polishing and editing. Specifically, we utilized its capabilities to improve grammar, refine sentence structure, enhance clarity, and ensure stylistic consistency throughout the paper. This process helped us articulate our ideas more effectively and adhere to the high standards of academic writing.

It is crucial to emphasize that all core scientific aspects of this research were conducted entirely by the human authors without the assistance of any LLM. This includes, but is not limited to:

- The conceptualization of the research problem and the formulation of the main hypotheses.
- The design of the model architecture, algorithms, and the overall methodological framework.
- The collection, preprocessing, and curation of all datasets used in our experiments.
- The writing and implementation of all source code for experiments and data analysis.
- The execution of all experiments, the generation of results, and the subsequent data analysis.
- The interpretation of the results and the formulation of the scientific conclusions and future work.

The intellectual contributions, technical innovations, and scientific insights presented in this paper are exclusively the work of the authors. We critically reviewed and edited all text generated or modified by the LLM to ensure it accurately reflects our original thoughts and findings. The authors bear full responsibility for the final content, scientific accuracy, and all claims made in this work.

## B  DERIVATION AND PROOF

### B.1  NOTATION AND SETUP

Let $E_\phi : \mathcal{X} \to \mathbb{R}$ be a differentiable scoring head with parameters $\phi$ (the encoder is frozen). For any ID–OOD pair $(S_{\text{in}}, S_{\text{out}})$ we define the margin

$$\Delta E_\phi \triangleq E_\phi(S_{\text{out}}) - E_\phi(S_{\text{in}}), \tag{15}$$

which is identical to Eq. 8 in the main text. The per-pair logistic term is

$$\ell(\Delta E_\phi) = \log\!\big(1 + \exp(-\beta\,\Delta E_\phi)\big), \tag{16}$$

and the objective is the pairwise expectation

$$\mathcal{L}(\phi) = \mathbb{E}_{S_{\text{in}}\sim D_{\text{in}},\, S_{\text{out}}\sim D_{\text{out}}}\big[\ell(\Delta E_\phi)\big], \tag{17}$$

which coincides with Eq. 6 (and its expectation form Eq. 14 after the change $f = -E_\phi$ used in the main text). When needed, we also consider the total objective with a small $\ell_2$ gauge-fixing term

$$\mathcal{L}_{\text{tot}}(\phi) = \mathcal{L}(\phi) + \frac{\lambda}{2}\,\mathbb{E}_{S\sim\Pi}\big[E_\phi(S)^2\big], \tag{18}$$

matching Eq. 7 in the main text; here $\Pi$ is any fixed sampling measure on $\mathcal{X}$ (e.g., the mixture of $D_{\text{in}}$ and $D_{\text{out}}$).

### B.2  FULL GRADIENT AND CURVATURE CALCULATIONS

**Gradient w.r.t. the margin.**   For $z = \Delta E_\phi$ we have

$$\ell'(z) = \frac{\partial}{\partial z}\log\!\big(1 + e^{-\beta z}\big) = -\frac{\beta}{1 + e^{\beta z}} = -\beta\,\sigma(-\beta z),$$

$$\ell''(z) = \frac{\partial}{\partial z}\ell'(z) = \beta^2\,\sigma(\beta z)\,\sigma(-\beta z) > 0,$$

so $\ell$ is strictly convex in $z$.

**Gradient w.r.t. parameters.** By chain rule,

$$\nabla_\phi \ell(\Delta E_\phi) = \ell'(\Delta E_\phi) \, \nabla_\phi \Delta E_\phi \tag{19}$$
$$= -\beta \, \sigma(-\beta \Delta E_\phi) \big(\nabla_\phi E_\phi(S_{\text{out}}) - \nabla_\phi E_\phi(S_{\text{in}})\big).$$

Define the direction term

$$d_\phi \triangleq \nabla_\phi E_\phi(S_{\text{out}}) - \nabla_\phi E_\phi(S_{\text{in}}),$$

and the weight function

$$w_\beta(t) \triangleq \beta \, \sigma(-\beta t) \in (0, \beta). \tag{20}$$

Then $\nabla_\phi \ell(\Delta E_\phi) = -w_\beta(\Delta E_\phi) \, d_\phi$ and

$$\nabla_\phi \mathcal{L}(\phi) = \mathbb{E}\big[-w_\beta(\Delta E_\phi) \, d_\phi\big].$$

Moreover,

$$w'_\beta(t) = -\beta^2 \sigma(\beta t)\sigma(-\beta t) < 0, \qquad w_\beta(0) = \beta/2, \quad \lim_{t \to -\infty} w_\beta(t) = \beta, \quad \lim_{t \to +\infty} w_\beta(t) = 0,$$

so misranked ($t < 0$) or borderline ($t \approx 0$) pairs receive larger weights, while already-separated pairs ($t \gg 0$) are nearly ignored.

**Per-example backprop signals.** Differentiating Eq. 16 w.r.t. the scalar ranking score gives

$$\frac{\partial \ell}{\partial E_\phi(S_{\text{out}})} = -\beta \, \sigma(-\beta \Delta E_\phi), \qquad \frac{\partial \ell}{\partial E_\phi(S_{\text{in}})} = +\beta \, \sigma(-\beta \Delta E_\phi),$$

equal in magnitude and opposite in sign. If $E_\phi$ is linear on a frozen representation $h(\cdot)$, i.e., $E_\phi(S) = \langle w, h(S) \rangle + b$, then $d_\phi$ reduces to $h(S_{\text{out}}) - h(S_{\text{in}})$ and the gradient is a simple contrastive update.

### B.3 PROOF OF PROPOSITION 4.1

We show that a gradient step increases the margin by an amount proportional to $w_\beta(\Delta E_\phi)\|d_\phi\|_2^2$.

**First-order calculation.** A gradient descent step with stepsize $\eta > 0$ is

$$\phi^+ = \phi - \eta \, \nabla_\phi \ell(\Delta E_\phi) = \phi + \eta \, w_\beta(\Delta E_\phi) \, d_\phi.$$

By first-order Taylor expansion,

$$\delta(\Delta E_\phi) \triangleq \Delta E_\phi(\phi^+) - \Delta E_\phi(\phi)$$
$$\approx \nabla_\phi \Delta E_\phi(\phi)^\top (\phi^+ - \phi)$$
$$= d_\phi^\top \big(\eta \, w_\beta(\Delta E_\phi) \, d_\phi\big) \tag{21}$$
$$= \eta \, w_\beta(\Delta E_\phi) \, \|d_\phi\|_2^2$$
$$\geq 0.$$

which yields Eq. 9 in the main text with $w_\beta(\cdot) = \beta\sigma(-\beta\cdot)$.

**Second-order control (sufficient condition for strict increase).** If $\nabla_\phi \Delta E_\phi$ is $L$-Lipschitz (equivalently, the Hessian of $\Delta E_\phi$ has operator norm $\leq L$ on the segment between $\phi$ and $\phi^+$), then Taylor's theorem gives

$$\delta(\Delta E_\phi) \geq \eta \, w_\beta(\Delta E_\phi) \, \|d_\phi\|_2^2 - \frac{L}{2} \big\|\eta \, w_\beta(\Delta E_\phi) \, d_\phi\big\|_2^2 = \eta \, w_\beta(\Delta E_\phi) \, \|d_\phi\|_2^2 \Big(1 - \tfrac{L\eta}{2} \, w_\beta(\Delta E_\phi)\Big).$$

Hence for any pair, choosing $\eta < 2/\big(L \, w_\beta(\Delta E_\phi)\big)$ guarantees $\delta(\Delta E_\phi) > 0$.

### B.4 PROOF OF LEMMA 4.2 (LOCAL PAIRWISE OPTIMALITY)

Define $f = -E_\phi$ so that larger $f$ is "more ID-like", as in the main text. For two samples $S, S'$ let $z = f(S) - f(S') = E_\phi(S') - E_\phi(S)$ (Eq. 11). Let $\eta(S) = \Pr(\text{ID} \mid S)$ and set

$$\alpha \triangleq \eta(S)\big(1 - \eta(S')\big), \qquad \alpha' \triangleq \eta(S')\big(1 - \eta(S)\big).$$

The conditional pairwise risk (Eq. 12) can be written

$$r_\beta(z; S, S') = \alpha \log(1 + e^{-\beta z}) + \alpha' \log(1 + e^{\beta z}).$$

*Strict convexity.* Using $\ell''(z) > 0$ from B.2, we have

$$\frac{\partial^2 r_\beta}{\partial z^2} = (\alpha + \alpha') \beta^2 \sigma(\beta z)\sigma(-\beta z) > 0 \quad \text{whenever } \alpha + \alpha' > 0,$$

so $r_\beta(z; S, S')$ is strictly convex in $z$ for any non-degenerate pair.

*Stationary point.* Differentiating and setting to zero,

$$\frac{\partial r_\beta}{\partial z} = -\alpha \beta \sigma(-\beta z) + \alpha'\beta \sigma(\beta z) = 0 \quad \Longleftrightarrow \quad \alpha \sigma(-\beta z) = \alpha' \sigma(\beta z).$$

Using $\sigma(\beta z)/\sigma(-\beta z) = e^{\beta z}$,

$$e^{\beta z^\star} = \frac{\alpha}{\alpha'} = \frac{\eta(S)\,[1 - \eta(S')]}{\eta(S')\,[1 - \eta(S)]} \quad \Longrightarrow \quad z^\star = \frac{1}{\beta} \log \frac{\eta(S)\,[1 - \eta(S')]}{\eta(S')\,[1 - \eta(S)]},$$

which is Eq. 13. Since $u \mapsto \log(u/(1 - u))$ is strictly increasing on $(0, 1)$, $\text{sign}(z^\star) = \text{sign}\big(\eta(S) - \eta(S')\big)$.

### B.5 PROOF OF PROPOSITION 4.3 (GLOBAL BAYES-OPTIMAL RANKING)

We now prove that, over a rich function class, any global minimizer of the population risk

$$\mathcal{R}_\beta(f) = \mathbb{E}_{(S_{\text{in}}, S_{\text{out}})}\Big[\log\big(1 + e^{-\beta\,[\,f(S_{\text{in}}) - f(S_{\text{out}})\,]}\big)\Big] \quad \text{(Eq. 14)}$$

induces the same ordering as $\eta(\cdot)$ for almost all pairs, hence achieves the Bayes-optimal AUROC.

**A canonical minimizer realizing the point-wise optima simultaneously.** Define

$$f^\star(S) \triangleq \frac{1}{\beta} \log \frac{\eta(S)}{1 - \eta(S)} \quad (+ \text{ any additive constant}). \tag{22}$$

For any pair $(S, S')$, $z^\star = f^\star(S) - f^\star(S')$ equals the two-point optimum from Lemma 4.2. Therefore, for every pair, $r_\beta\big(f^\star(S) - f^\star(S'); S, S'\big)$ attains the *pairwise* minimal value. Integrating over pairs shows $f^\star$ minimizes the population risk $\mathcal{R}_\beta$. Moreover, if $g$ is any other function with $\mathcal{R}_\beta(g) = \mathcal{R}_\beta(f^\star)$, then for almost every pair $(S, S')$ we must have $g(S) - g(S') = f^\star(S) - f^\star(S')$, hence $g - f^\star$ is almost everywhere constant. Thus the set of global minimizers is exactly $\{ f^\star + c : c \in \mathbb{R} \}$. In particular, all global minimizers induce the same ordering as $\eta$.

**Remarks.** (i) Any strictly increasing transform $h \circ \eta$ induces the same ranking and hence achieves Bayes-optimal AUROC; however, it does not in general minimize $\mathcal{R}_\beta$ unless $h(u) = \beta^{-1} \log\big(u/(1 - u)\big)$ up to an additive constant, because only then do all pairwise gaps equal $z^\star$. (ii) The additive constant does not affect pairwise differences and hence leaves both AUROC and $\mathcal{R}_\beta$ unchanged; the $\ell_2$ term in Eq. 18 fixes this gauge without changing the induced ranking.

### B.6 AUROC CONNECTION AND TEMPERATURE LIMIT

AUROC may be written as $\Pr\big(E_\phi(S_{\text{in}}) < E_\phi(S_{\text{out}})\big)$, i.e., the probability that ID ranks ahead of OOD (equivalently, $\Pr(\Delta E_\phi > 0)$), which is exactly the ranking event smoothed by $\ell(\cdot)$ in equation 16. As $\beta \to \infty$,

$$\ell(z) = \log(1 + e^{-\beta z}) \longrightarrow \begin{cases} 0, & z > 0, \\ \log 2, & z = 0, \\ +\infty, & z < 0, \end{cases}$$

so the function converges to a hard 0–1 ranking penalty that forbids misordered pairs ($z < 0$).

### B.7 ADDITIONAL IMPLEMENTATION-FACING IDENTITIES

**Steepness around the boundary.** The magnitude of the per-pair backprop signal is

$$\left\|\nabla_\phi \ell(\Delta E_\phi)\right\| = w_\beta(\Delta E_\phi)\|d_\phi\| \leq \frac{\beta}{2}\|d_\phi\| \quad \text{with equality at } \Delta E_\phi = 0.$$

Hence updates concentrate near the decision boundary; the logistic slope is maximized at $\Delta E_\phi = 0$.

**Invariance to shifts: role of $\ell_2$ regularization.** For any constant $c$, replacing $E_\phi$ by $E_\phi + c$ leaves $\Delta E_\phi$ (and hence $\mathcal{L}(\phi)$ and AUROC) unchanged. The $\ell_2$ term in Eq. 18 removes this degree of freedom by selecting the unique representative (up to sampling) with minimal squared energy norm, without affecting the ranking.

**Effect of temperature $\beta$.** Larger $\beta$ sharpens $w_\beta(t)$ towards a hard 0–1 ranking loss, leading to faster correction of borderline pairs but potentially larger variance if margins become too large (gradient saturation for well-separated pairs). Smaller $\beta$ smooths updates and can be numerically more forgiving; in all cases, Proposition 4.3 ensures the same Bayes-optimal ranking at the population optimum.

## C EXPERIMENTS DETAILS

### C.1 DATASET

**DrugOOD Dataset.** DrugOOD (Ji et al., 2022) is a benchmark and automated dataset curator for OOD challenges in AI-aided drug discovery, built from the large-scale ChEMBL bioassay database. It focuses on the crucial task of drug-target binding affinity prediction, which is framed as a binary classification problem (active vs. inactive). For this, we focus on the Ligand-Based Affinity Prediction (LBAP) variants of DrugOOD, which define three types of domain shifts based on biochemistry knowledge:

- **Assay**: Samples are split by the experimental assay, simulating shifts in experimental environments. Assays with many samples are used for training, while those with fewer are used for testing.
- **Scaffold**: Samples are split by their molecular scaffold structure. The largest scaffolds are assigned to the training set and the smallest to the test set to maximize structural diversity.
- **Size**: Samples are split by the number of atoms in the molecule. Molecules with the largest atomic sizes are used for training and smaller ones for testing to ensure variability.

Specifically, we use the `drugood_lbap_general_[ec50, ic50]_(assay, scaffold, size)` subsets, which are the standard LBAP partitions provided in DrugOOD. These subsets cover different types of domain shifts under both `EC50` and `IC50` measurement settings, offering diverse and challenging scenarios for OOD evaluation. Deterministic splits are constructed using `data_seed=42`, with the following target sizes: `train_id=2000`, `train_ood=2000`, `val_id=600`, `val_ood=600`, `test_id=1000`, and `test_ood=1000`. Supervised training labels are read from the `cls_label` field in the JSON files.

Table 3: List of used DrugOOD dataset. Pos and Neg denote the numbers of positive and negative data points, respectively. $D^\#$ represents the number of domains, and $C^\#$ represent the number of data points.

| Data subset | $Pos^\#$ | $Neg^\#$ | Train | | ID Val | | ID Test | | OOD Val | | OOD Test | |
|---|---|---|---|---|---|---|---|---|---|---|---|---|
| | | | $D^\#$ | $C^\#$ | $D^\#$ | $C^\#$ | $D^\#$ | $C^\#$ | $D^\#$ | $C^\#$ | $D^\#$ | $C^\#$ |
| drugood-lbap-general-ec50-assay | 92445 | 18000 | 1079 | 39333 | 1079 | 12849 | 1079 | 14086 | 1137 | 22095 | 2883 | 22082 |
| drugood-lbap-general-ec50-scaffold | 92445 | 18000 | 8677 | 23481 | 2450 | 4977 | 30611 | 37811 | 7659 | 22095 | 4844 | 22081 |
| drugood-lbap-general-ec50-size | 92445 | 18000 | 294 | 42697 | 238 | 14151 | 312 | 14531 | 4 | 19301 | 20 | 19765 |
| drugood-lbap-general-ic50-assay | 476865 | 91691 | 6917 | 201951 | 6917 | 65424 | 6917 | 73777 | 6207 | 113704 | 16814 | 113700 |
| drugood-lbap-general-ic50-scaffold | 476865 | 91691 | 43552 | 129740 | 13516 | 29174 | 142173 | 182220 | 29513 | 113723 | 15189 | 113699 |
| drugood-lbap-general-ic50-size | 476865 | 91691 | 290 | 217294 | 243 | 72349 | 311 | 72742 | 4 | 102544 | 22 | 103627 |

**GOOD Datasets.** The GOOD (Gui et al., 2022) benchmark provides molecular graph datasets designed for systematic OOD evaluation. We use three datasets:

- **GOOD-HIV**: A small-scale dataset adapted from MoleculeNet. Inputs are molecular graphs with atoms as nodes and chemical bonds as edges. The task is binary classification to predict whether a molecule inhibits HIV replication.
- **GOOD-PCBA**: A large-scale dataset with 128 bioassays, forming a multi-target binary classification task.
- **GOOD-ZINC**: A molecular property regression dataset derived from the ZINC database, where molecules contain up to 38 heavy atoms. The task is to predict constrained solubility.

For our experiments, we generate deterministic splits with sizes: `train_id/train_ood=5000`, `val_id/val_ood=1500`, and `test_id/test_ood=2000`. Labels strictly follow the official benchmark protocol. For tasks that are regression-like, such as `GOOD-ZINC`, we binarize the target via a median split: samples with values greater than or equal to the median are assigned positive labels, and the rest are negative. This allows for the training of a supervised classifier, and OOD detection is then evaluated on the classifier's outputs. Statistics of the datasets are summarized in Table 4.

**Splitting, caching, and features.** All dataset splits are generated deterministically using seed 42. We pre-compute molecular representations using foundation encoders and cache them for reuse. Feature extraction is performed with batch size 50, and the resulting representations are used both to train the supervised classifier head and as the basis for OOD scoring methods.

## C.2 BASELINES

We compare against several widely used supervised OOD detection baselines, all of which operate on the logits or penultimate features of a trained classifier head. Unless otherwise specified, the OOD score is defined such that larger values indicate a higher likelihood of being out-of-distribution.

**MSP** (Hendrycks & Gimpel, 2016) calculates the in-distribution (ID) score as the maximum class confidence from the classifier head's logits. For multi-class tasks, this is the maximum softmax probability, while for binary or single-output tasks it is $\max(p, 1-p)$ from the sigmoid output. In multi-task settings, the confidence is first computed for each task and then averaged to yield the final ID score. The reported OOD score is defined as $1 - $ ID score.

**Energy** (Liu et al., 2020) uses the energy score defined as $E(x) = -\log \sum_c \exp(z_c)$, where $z$ are the logits. For binary or single-output tasks, we augment the logits as $[z, -z]$ before computing the log-sum-exp. In the multi-task case, the energy is computed per task and then averaged. Larger energy values correspond to higher OOD likelihood.

**ODIN** (Liang et al., 2017) is implemented in our setting directly on the pre-computed molecular feature representations that serve as inputs to the classifier head, rather than on raw molecules. A small perturbation is applied according to $x' = x - \varepsilon \cdot \text{sign}(\partial \mathcal{L}_{\text{CE}}/\partial x)$, with $\varepsilon = 0.0014$. Temperature scaling with $T = 1000$ is applied exactly once to the logits. The model is kept in evaluation mode with Batch Normalization layers frozen. The confidence score is computed in the same way as MSP after perturbation, and task-level confidences are averaged in the multi-task case. The OOD score is then reported as $1 - $ confidence.

**Mahalanobis** (Lee et al., 2018) operates in the penultimate feature space of the classifier head. For single-task classification, it estimates class-conditional means and a shared precision matrix, where Ledoit–Wolf shrinkage is used preferentially and empirical covariance is used as a fallback. The OOD score for a test sample is its minimum Mahalanobis distance to any class mean. In the multi-task setting, we instead fit a single global mean and a shared precision matrix, and the OOD score is given by the Mahalanobis distance to this global mean. Larger distances indicate higher OOD likelihood.

**KNN** (Sun et al., 2022) also works in the penultimate feature space, standardized using the mean and standard deviation of the training set. The OOD score is defined as the average Euclidean distance to the $k = 50$ nearest neighbors in this space. A larger average distance indicates that the sample is more likely to be OOD.

| Dataset | Shift | Train | ID validation | ID test | OOD validation | OOD test | Train | OOD validation | ID validation | ID test | OOD test |
|---------|-------|-------|---------------|---------|----------------|---------|-------|----------------|---------------|---------|---------|
| | | | | Scaffold | | | | | Size | | |
| GOOD-HIV | covariate | 24682 | 4112 | 4112 | 4113 | 4108 | 26169 | 4112 | 4112 | 2773 | 3961 |
| | concept | 15209 | 3258 | 3258 | 9365 | 10037 | 14454 | 3096 | 3096 | 9956 | 10525 |
| | no shift | 24676 | 8225 | 8226 | - | - | 24676 | 8225 | 8226 | - | - |
| | | | | Scaffold | | | | | Size | | |
| GOOD-PCBA | covariate | 262764 | 43792 | 43792 | 44019 | 43562 | 269990 | 43792 | 43792 | 48430 | 31925 |
| | concept | 159158 | 34105 | 34105 | 90740 | 119821 | 150121 | 32168 | 32168 | 108267 | 115205 |
| | no shift | 262757 | 87586 | 87586 | - | - | 262757 | 87586 | 87586 | - | - |
| | | | | Scaffold | | | | | Size | | |
| GOOD-ZINC | covariate | 149674 | 24945 | 24945 | 24945 | 24946 | 161893 | 24945 | 24945 | 20270 | 17402 |
| | concept | 101867 | 21828 | 21828 | 43539 | 60393 | 89418 | 19161 | 19161 | 51409 | 70306 |
| | no shift | 149673 | 49891 | 49891 | - | - | 149673 | 49891 | 49891 | - | - |

Table 4: Numbers of graphs in training, ID validation, ID test, OOD validation, and OOD test sets for the GOOD datasets.

**LOF** (Breunig et al., 2000) is applied in the same standardized feature space. A Local Outlier Factor (LOF) model is fitted in novelty detection mode with $n_{\text{neighbors}} = 20$. The OOD score is defined as $-\text{score\_samples}(\cdot)$, so that higher values correspond to samples that deviate more strongly from the local density of their neighbors.

**Monte Carlo dropout** (Gal & Ghahramani, 2016) approximates Bayesian inference by maintaining stochastic dropout during the inference phase. In our implementation, the model is kept in evaluation mode to freeze Batch Normalizations statistics, while dropout layers are explicitly set to trainging mode. We perform $T = 20$ stochastic forward passes for each test input. The raw logits from each pass are converted to probabilities—using softmax for multi-class tasks or sigmoid for binary tasks—and then averaged to obtain the expected predictive distribution. The OOD score is defined as the predictive entropy of this mean distribution. Higher entropy indicates higher uncertainty, corresponding to a higher likelihood of being OOD

**Conformal Prediction** (Angelopoulos & Bates, 2021) adopts a standard split conformal prediction, utilizing the full validation set as the calibration set $\mathcal{C}$. The conformity score $s(x)$ is defined as the maximum predicted probability (averaged across tasks for multi-task settings). For a test input $x$, we compute the conformal p-value relative to the calibration scores: $p(x) = \frac{1 + \sum_{s_i \in \mathcal{C}} \mathbb{I}(s_i \leq s(x))}{1 + |\mathcal{C}|}$. The OOD score is reported as $1 - p(x)$, such that lower p-values indicate a higher likelihood of being out-of-distribution.

## D ADDITIONAL EXPERIMENTAL RESULTS

We supplement additional experimental results in this section to provide a comprehensive evaluation of Mole-PAIR. Specifically, we visualize the gradient weight analysis in Figure 6 to empirically verify the self-paced learning behavior. To demonstrate robustness, we report the impact of OOD data scarcity in Table 5 and the generalization ability across different assays and splits in Table 6. Furthermore, we compare Mole-PAIR with state-of-the-art domain-specific graph OOD methods in Table 7 and uncertainty-based baselines in Table 8. Finally, we present the computational efficiency comparison in Table 9 and an ablation study regarding backbone fine-tuning strategies in Table 10.

### D.1 GRADIENT WEIGHT ANALYSIS

**Prioritization of Difficult Pairs.** Across all shifts we observe the theoretically predicted ordering *Hard > Boundary > Easy* (Figure 6), with the boundary mean essentially equal to $w_\beta(0) = 0.05$. These ratios quantify the self-paced behavior predicted by Proposition 4.1: updates concentrate on misranked or borderline pairs (larger $w_\beta$) and spend little budget on already well-separated pairs (smaller $w_\beta$). The exact boundary level of 0.0500 further validates the loss in Eq. 6 and its gradient weighting in Eq. 20.

**Shift-Specific Weight Distribution.** Assay shows the smallest hard-over-easy advantage (+14.7%), indicating that many misranked pairs are only mildly negative, while easy pairs are not

extremely far from the boundary. Scaffold exhibits a larger advantage (+28.4%), consistent with more well-separated easy pairs and hard pairs concentrated near the boundary. Size achieves the largest advantage (+32.4%) because easy pairs are very far from the boundary, whereas the remaining hard pairs are only slightly negative. This ordering mirrors the training dynamics and the $\beta/\lambda$ sensitivity observed in the main text: Size splits are geometrically easy and quickly cleaned up; Scaffold benefits from a stronger preference signal; and Assay improves more gradually.

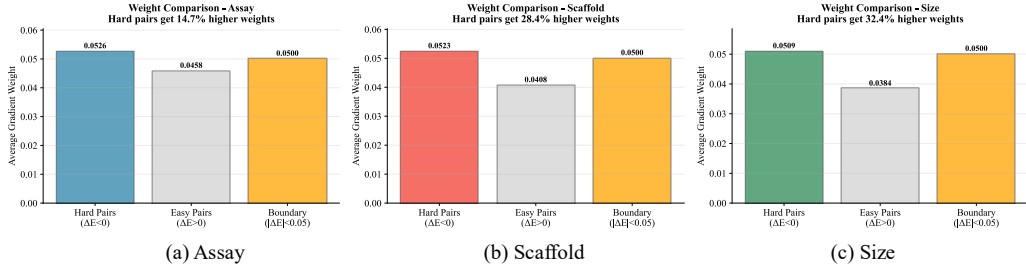

|                   |                   |                   |
|:-----------------:|:-----------------:|:-----------------:|
| (a) Assay         | (b) Scaffold      | (c) Size          |

Figure 6: **Hard vs. Easy vs. Boundary: average gradient weights across shifts.** Bars show the mean of the per-pair update weight $w_\beta(\Delta) = \beta\,\sigma(-\beta\Delta)$ under three EC50 shifts (Assay, Scaffold, Size) with $\beta = 0.1$. Groups are: Hard pairs ($\Delta E_\phi < 0$), Easy pairs ($\Delta E_\phi > 0$), and Boundary pairs ($|\Delta E_\phi| < 0.05$).

**Implications for AUROC Optimization.** Because AUROC equals $\Pr\big(E_\phi(S_{\mathrm{ID}}) < E_\phi(S_{\mathrm{OOD}})\big)$, the larger $w_\beta$ on hard or borderline pairs ensures that training first reduces the mass of misordered pairs and then widens margins. This mechanism explains the down→down→up trajectories in Figure 5 and the strong test-time gains reported in Table 1 and Table 2.

### D.2 IMPACT OF OOD DATA SCARCITY

To validate the efficacy of Mole-PAIR under extreme OOD data scarcity, we manipulated the ID:OOD ratio in the training data. Specifically, we reduced the number of training OOD samples while keeping the number of ID samples, as well as the validation and test splits, constant. We varied the training ID:OOD ratio from 1:1 to 1:0.01 to simulate increasingly scarce OOD availability. The frozen foundation model used here is MiniMol. The AUROC results for Mole-PAIR are shown below:

Table 5: Performance of Mole-PAIR on DrugOOD datasets with different ID:OOD ratios

| ID:OOD Ratio         | 1:1               | 1:0.1             | 1:0.02            | 1:0.01            |
|----------------------|-------------------|-------------------|-------------------|-------------------|
| DrugOOD-EC50-Scaffold | $0.970 \pm 0.000$ | $0.924 \pm 0.001$ | $0.898 \pm 0.003$ | $0.849 \pm 0.005$ |
| DrugOOD-EC50-Size     | $1.000 \pm 0.000$ | $0.993 \pm 0.001$ | $0.991 \pm 0.001$ | $0.974 \pm 0.020$ |
| DrugOOD-EC50-Assay    | $0.711 \pm 0.002$ | $0.638 \pm 0.005$ | $0.621 \pm 0.004$ | $0.603 \pm 0.002$ |

We observed that performance degradation was moderate even when training OOD samples became extremely scarce. For example, at the 1:0.01 ratio, there are only 20 training OOD samples. However, even under such severe data scarcity, our model still outperformed the strongest baseline trained with the full 1:1 ratio. These results serve as strong empirical evidence that Mole-PAIR is robust to limited and imbalanced OOD availability, remaining a cost-effective post-training remedy even with very few OOD samples.

### D.3 GENERALIZATION ABILITY

To fully evaluate the generalization ability of our proposed Mole-PAIR, we train and validate Mole-PAIR on DrugOOD-EC50-Scaffold, and test it on DrugOOD-IC50-Size. In addition, we also eval-

uate the reverse setting by training on the Size split and testing on the Scaffold split to ensure bidirectional robustness. The AUROC results are shown below:

Table 6: Generalization performance of Mole-PAIR across different assay types and splitting strategies

| Training Set | Test Set | AUROC | AUPR | FPR95 |
|---|---|---|---|---|
| DrugOOD-EC50-Scaffold | DrugOOD-IC50-Size | $0.968 \pm 0.001$ | $0.973 \pm 0.001$ | $0.196 \pm 0.003$ |
| DrugOOD-EC50-Size | DrugOOD-EC50-Size | $1.000 \pm 0.000$ | $1.000 \pm 0.000$ | $0.000 \pm 0.000$ |
| DrugOOD-IC50-Scaffold | DrugOOD-EC50-Size | $0.984 \pm 0.001$ | $0.986 \pm 0.001$ | $0.069 \pm 0.003$ |
| DrugOOD-IC50-Size | DrugOOD-IC50-Size | $0.999 \pm 0.000$ | $0.999 \pm 0.001$ | $0.004 \pm 0.001$ |

The results highlight the exceptional generalization ability of Mole-PAIR. We observe that the model achieves consistent high performance not only on matched datasets but also on completely unseen distributions involving different assays and splitting criteria. Specifically, the successful transfer from Scaffold-based training to Size-based testing (and vice versa) suggests that our model effectively learns invariant features that are robust to severe distribution shifts. This strong empirical evidence verifies that Mole-PAIR is not merely memorizing training data patterns but has acquired a generalized understanding of molecular OOD characteristics.

### D.4 COMPARISON WITH DOMAIN-SPECIFIC GRAPH OOD APPROACHES

To further validate the effectiveness of Mole-PAIR, we compare our models with domain-specific OOD methods, which typically adopt different architectures or require complex generative training. The AUROC results are shown in Table 7.

Table 7: Comparison with state-of-the-art OOD detection methods on DrugOOD datasets measured by **AUROC** (↑). The best results are indicated in boldface, and the second best results are underlined.

| Method | DrugOOD-EC50 | | DrugOOD-IC50 | |
|---|---|---|---|---|
| | Scaffold | Size | Scaffold | Size |
| GraphDE | $0.686 \pm 0.010$ | $0.796 \pm 0.012$ | $0.692 \pm 0.011$ | $0.787 \pm 0.010$ |
| GOOD-D | $0.825 \pm 0.013$ | $0.925 \pm 0.013$ | $0.854 \pm 0.012$ | $0.916 \pm 0.011$ |
| PGR-MOOD | $\underline{0.875 \pm 0.013}$ | $\underline{0.977 \pm 0.015}$ | $\underline{0.916 \pm 0.013}$ | $\underline{0.938 \pm 0.015}$ |
| Mole-PAIR | $\mathbf{0.970 \pm 0.000}$ | $\mathbf{1.000 \pm 0.000}$ | $\mathbf{0.983 \pm 0.000}$ | $\mathbf{0.999 \pm 0.000}$ |

The results show that Mole-PAIR consistently outperforms these dedicated molecular or graph OOD methods, surpassing the recent SOTA method PGR-MOOD (Shen et al., 2024) by significant margins across different splits. Unlike these domain-specific models that require complex generative training with specific architectures, Mole-PAIR achieves superior performance using a simple ranking objective that only trains a lightweight scoring head, demonstrating the superiority of the method.

Table 8: Comparison with uncertainty-based methods on DrugOOD and Good-HIV datasets measured by **AUC** (↑). The best results are indicated in **boldface**, and the second best results are underlined.

| Method | DrugOOD-EC50-Scaffold | Good-HIV-Scaffold |
|---|---|---|
| MSP (Baseline) | $0.677 \pm 0.093$ | $0.408 \pm 0.054$ |
| MC Dropout | $0.728 \pm 0.015$ | $0.538 \pm 0.055$ |
| Conformal Prediction | $\underline{0.803 \pm 0.007}$ | $\underline{0.576 \pm 0.074}$ |
| Mole-PAIR (Ours) | $\mathbf{0.970 \pm 0.000}$ | $\mathbf{0.777 \pm 0.003}$ |

Table 9: Computational efficiency comparison measured by Training Time (s) and GPU Memory (GB). Lower values are better (↓). The best results are indicated in **boldface**, and the second best results are underlined.

| Method | Training Time (s) ↓ | GPU Memory (GB) ↓ |
|---|---|---|
| MSP | 108.09 | **1.52** |
| ODIN | 108.35 | 1.53 |
| Energy | **107.12** | **1.52** |
| Mahalanobis | 108.31 | **1.52** |
| LOF | 108.43 | **1.52** |
| KNN | 117.47 | **1.52** |
| Mole-PAIR (Ours) | 112.61 | 1.60 |

Table 10: Ablation study on different fine-tuning strategies of Uni-Mol backbone measured by **AUC** (↑). The best results are indicated in **boldface**, and the second best results are underlined.

| Fine-tuning Strategy | DrugOOD-EC50-Scaffold | DrugOOD-EC50-Assay |
|---|---|---|
| Full fine-tuning (Uni-Mol) | 0.833 | 0.583 |
| Partial fine-tuning (Uni-Mol) | 0.943 | 0.623 |
| Frozen (Uni-Mol) | **0.965** | **0.650** |

Table 11: Model performance comparison: out-of-distribution detection results are measured by **AUROC** (↑) / **AUPR** (↑) / **FPR95** (↓).

| | Metrics | MiniMol | | | | | | | Uni-mol | | | | | | |
|---|---|---|---|---|---|---|---|---|---|---|---|---|---|---|---|
| | | MSP | ODIN | Energy | Mah | LOF | KNN | Mole-PAIR | MSP | ODIN | Energy | Mah | LOF | KNN | Mole-PAIR |
| EC50-Scaffold | AUROC | 0.677±0.093 | 0.633±0.154 | 0.685±0.104 | 0.660±0.047 | 0.665±0.033 | 0.671±0.052 | **0.970**±0.000 | 0.694±0.100 | 0.621±0.195 | 0.691±0.091 | 0.724±0.044 | 0.748±0.033 | 0.704±0.040 | **0.965**±0.001 |
| | AUPR | 0.627±0.081 | 0.597±0.108 | 0.625±0.096 | 0.696±0.043 | 0.690±0.026 | 0.700±0.050 | **0.975**±0.000 | 0.692±0.099 | 0.620±0.146 | 0.681±0.086 | 0.717±0.061 | 0.763±0.034 | 0.696±0.050 | **0.972**±0.001 |
| | FPR95 | 0.725±0.109 | 0.734±0.128 | 0.716±0.111 | 0.898±0.023 | 0.893±0.025 | 0.870±0.042 | **0.178**±0.010 | 0.848±0.070 | 0.816±0.135 | 0.849±0.083 | 0.831±0.033 | 0.844±0.038 | 0.855±0.018 | **0.178**±0.008 |
| EC50-Size | AUROC | 0.449±0.149 | 0.450±0.149 | 0.527±0.281 | 0.831±0.059 | 0.838±0.024 | 0.855±0.047 | **1.000**±0.000 | 0.722±0.112 | 0.656±0.171 | 0.652±0.178 | 0.784±0.064 | 0.855±0.048 | 0.747±0.062 | **1.000**±0.000 |
| | AUPR | 0.453±0.095 | 0.453±0.096 | 0.541±0.204 | 0.836±0.055 | 0.847±0.022 | 0.853±0.039 | **1.000**±0.000 | 0.697±0.116 | 0.652±0.131 | 0.635±0.157 | 0.726±0.065 | 0.849±0.055 | 0.691±0.064 | **1.000**±0.000 |
| | FPR95 | 0.833±0.110 | 0.832±0.011 | 0.726±0.293 | 0.633±0.115 | 0.667±0.081 | 0.550±0.154 | **0.000**±0.000 | 0.733±0.167 | 0.798±0.153 | 0.740±0.196 | 0.605±0.097 | 0.554±0.106 | 0.707±0.070 | **0.000**±0.000 |
| EC50-Assay | AUROC | 0.420±0.011 | 0.437±0.038 | 0.455±0.042 | 0.572±0.008 | 0.537±0.011 | 0.569±0.010 | **0.711**±0.002 | 0.547±0.038 | 0.558±0.061 | 0.547±0.037 | 0.591±0.017 | 0.558±0.016 | 0.599±0.022 | **0.650**±0.004 |
| | AUPR | 0.445±0.012 | 0.454±0.021 | 0.480±0.030 | 0.577±0.007 | 0.545±0.008 | 0.571±0.007 | **0.698**±0.002 | 0.555±0.028 | 0.561±0.039 | 0.553±0.028 | 0.590±0.014 | 0.569±0.013 | 0.595±0.016 | **0.641**±0.005 |
| | FPR95 | 0.972±0.002 | 0.950±0.063 | 0.973±0.004 | 0.929±0.010 | 0.933±0.011 | 0.930±0.010 | **0.823**±0.007 | 0.957±0.021 | 0.913±0.096 | 0.948±0.010 | 0.921±0.011 | 0.935±0.010 | 0.914±0.012 | **0.869**±0.009 |
| IC50-Scaffold | AUROC | 0.574±0.158 | 0.575±0.158 | 0.562±0.166 | 0.620±0.063 | 0.625±0.034 | 0.655±0.051 | **0.983**±0.000 | 0.585±0.055 | 0.546±0.048 | 0.542±0.059 | 0.699±0.024 | 0.717±0.053 | 0.682±0.047 | **0.977**±0.001 |
| | AUPR | 0.538±0.114 | 0.714±0.139 | 0.557±0.149 | 0.656±0.057 | 0.652±0.037 | 0.679±0.054 | **0.984**±0.000 | 0.604±0.048 | 0.571±0.048 | 0.561±0.057 | 0.667±0.033 | 0.714±0.055 | 0.654±0.047 | **0.980**±0.001 |
| | FPR95 | 0.759±0.126 | 0.563±0.257 | 0.781±0.147 | 0.909±0.037 | 0.909±0.013 | 0.898±0.010 | **0.084**±0.000 | 0.926±0.034 | 0.945±0.009 | 0.935±0.035 | 0.818±0.043 | 0.832±0.046 | 0.833±0.049 | **0.139**±0.005 |
| IC50-Size | AUROC | 0.515±0.219 | 0.516±0.220 | 0.573±0.325 | 0.751±0.060 | 0.745±0.070 | 0.765±0.064 | **0.999**±0.000 | 0.637±0.136 | 0.591±0.089 | 0.598±0.087 | 0.723±0.052 | 0.759±0.081 | 0.660±0.072 | **1.000**±0.000 |
| | AUPR | 0.494±0.119 | 0.495±0.119 | 0.593±0.250 | 0.765±0.061 | 0.753±0.071 | 0.776±0.063 | **0.999**±0.000 | 0.642±0.142 | 0.625±0.075 | 0.593±0.088 | 0.683±0.059 | 0.757±0.076 | 0.615±0.071 | **1.000**±0.000 |
| | FPR95 | 0.743±0.192 | 0.741±0.193 | 0.662±0.265 | 0.786±0.073 | 0.803±0.068 | 0.782±0.084 | **0.004**±0.001 | 0.861±0.137 | 0.911±0.051 | 0.872±0.073 | 0.718±0.077 | 0.746±0.106 | 0.806±0.080 | **0.000**±0.000 |
| IC50-Assay | AUROC | 0.600±0.018 | 0.614±0.007 | 0.569±0.053 | 0.516±0.023 | 0.564±0.010 | 0.502±0.031 | **0.660**±0.003 | 0.549±0.025 | 0.538±0.034 | 0.551±0.027 | 0.562±0.011 | 0.547±0.010 | 0.575±0.010 | **0.640**±0.003 |
| | AUPR | 0.580±0.007 | 0.590±0.011 | 0.550±0.052 | 0.520±0.017 | 0.561±0.004 | 0.504±0.023 | **0.653**±0.003 | 0.562±0.015 | 0.545±0.032 | 0.560±0.021 | 0.568±0.012 | 0.555±0.010 | 0.581±0.010 | **0.648**±0.004 |
| | FPR95 | 0.903±0.004 | 0.900±0.005 | 0.905±0.005 | 0.946±0.012 | 0.934±0.009 | 0.942±0.017 | **0.861**±0.010 | 0.945±0.022 | 0.943±0.014 | 0.949±0.023 | 0.936±0.006 | 0.942±0.007 | 0.929±0.011 | **0.890**±0.004 |
| HIV-Scaffold | AUROC | 0.408±0.054 | 0.390±0.057 | 0.388±0.037 | 0.503±0.037 | 0.508±0.024 | 0.562±0.066 | **0.777**±0.003 | 0.465±0.113 | 0.451±0.099 | 0.476±0.122 | 0.567±0.013 | 0.594±0.015 | 0.580±0.025 | **0.728**±0.004 |
| | AUPR | 0.438±0.034 | 0.430±0.037 | 0.423±0.040 | 0.497±0.030 | 0.499±0.018 | 0.538±0.049 | **0.740**±0.003 | 0.489±0.069 | 0.474±0.081 | 0.492±0.097 | 0.547±0.015 | 0.547±0.047 | 0.550±0.024 | **0.708**±0.003 |
| | FPR95 | 0.953±0.022 | 0.962±0.026 | 0.955±0.017 | 0.947±0.016 | 0.943±0.014 | 0.906±0.035 | **0.624**±0.008 | 0.954±0.018 | 0.947±0.022 | 0.952±0.028 | 0.920±0.017 | 0.894±0.021 | 0.885±0.021 | **0.736**±0.011 |
| HIV-Size | AUROC | 0.194±0.136 | 0.336±0.317 | 0.167±0.122 | 0.918±0.059 | 0.889±0.034 | 0.921±0.055 | **1.000**±0.000 | 0.218±0.081 | 0.278±0.116 | 0.243±0.154 | 0.826±0.035 | 0.860±0.038 | 0.826±0.026 | **1.000**±0.000 |
| | AUPR | 0.353±0.049 | 0.430±0.146 | 0.344±0.044 | 0.906±0.071 | 0.877±0.041 | 0.898±0.071 | **1.000**±0.000 | 0.353±0.023 | 0.387±0.062 | 0.376±0.093 | 0.801±0.040 | 0.862±0.036 | 0.806±0.036 | **1.000**±0.000 |
| | FPR95 | 0.983±0.032 | 0.819±0.243 | 0.985±0.029 | 0.315±0.144 | 0.458±0.084 | 0.280±0.130 | **0.001**±0.000 | 0.985±0.007 | 0.977±0.006 | 0.982±0.027 | 0.567±0.113 | 0.514±0.072 | 0.563±0.061 | **0.000**±0.000 |
| PCBA-Scaffold | AUROC | 0.632±0.037 | 0.623±0.032 | 0.642±0.034 | 0.564±0.038 | 0.564±0.030 | 0.459±0.078 | **0.924**±0.002 | 0.476±0.141 | 0.445±0.142 | 0.476±0.141 | 0.568±0.054 | 0.538±0.051 | 0.553±0.060 | **0.875**±0.001 |
| | AUPR | 0.601±0.036 | 0.591±0.034 | 0.615±0.027 | 0.559±0.032 | 0.575±0.035 | 0.485±0.060 | **0.924**±0.002 | 0.504±0.094 | 0.481±0.096 | 0.504±0.094 | 0.558±0.048 | 0.546±0.052 | 0.552±0.052 | **0.876**±0.001 |
| | FPR95 | 0.833±0.054 | 0.856±0.043 | 0.832±0.053 | 0.922±0.026 | 0.934±0.012 | 0.960±0.015 | **0.348**±0.011 | 0.954±0.050 | 0.962±0.054 | 0.954±0.050 | 0.836±0.294 | 0.948±0.026 | 0.941±0.031 | **0.515**±0.009 |
| PCBA-Size | AUROC | 0.697±0.073 | 0.691±0.056 | 0.735±0.091 | 0.728±0.119 | 0.687±0.156 | 0.527±0.160 | **1.000**±0.000 | 0.350±0.256 | 0.353±0.253 | 0.350±0.256 | 0.690±0.090 | 0.607±0.096 | 0.668±0.076 | **1.000**±0.000 |
| | AUPR | 0.654±0.087 | 0.657±0.084 | 0.700±0.117 | 0.713±0.129 | 0.696±0.147 | 0.519±0.103 | **1.000**±0.000 | 0.455±0.168 | 0.456±0.167 | 0.455±0.167 | 0.689±0.014 | 0.639±0.100 | 0.684±0.044 | **1.000**±0.000 |
| | FPR95 | 0.698±0.134 | 0.763±0.068 | 0.697±0.134 | 0.752±0.137 | 0.804±0.158 | 0.902±0.128 | **0.001**±0.000 | 0.925±0.157 | 0.925±0.157 | 0.925±0.157 | 0.855±0.066 | 0.926±0.037 | 0.895±0.039 | **0.000**±0.000 |
| ZINC-Scaffold | AUROC | 0.359±0.004 | 0.360±0.005 | 0.359±0.004 | **0.638**±0.011 | 0.544±0.056 | 0.636±0.028 | 0.614±0.004 | 0.352±0.114 | 0.352±0.114 | 0.352±0.114 | 0.643±0.113 | 0.577±0.041 | **0.645**±0.093 | 0.549±0.004 |
| | AUPR | 0.416±0.004 | 0.416±0.004 | 0.416±0.004 | **0.637**±0.010 | 0.548±0.057 | 0.637±0.041 | 0.587±0.003 | 0.437±0.099 | 0.437±0.098 | 0.437±0.098 | **0.694**±0.101 | 0.610±0.047 | 0.680±0.093 | 0.563±0.006 |
| | FPR95 | 0.985±0.003 | 0.985±0.003 | 0.985±0.003 | **0.905**±0.023 | 0.933±0.020 | 0.906±0.017 | 0.925±0.005 | 0.992±0.001 | 0.993±0.001 | 0.993±0.001 | 0.965±0.046 | **0.945**±0.018 | 0.961±0.042 | 0.949±0.002 |
| ZINC-Size | AUROC | 0.398±0.032 | 0.398±0.031 | 0.398±0.032 | 0.593±0.032 | 0.688±0.118 | 0.676±0.093 | **1.000**±0.000 | 0.369±0.058 | 0.369±0.058 | 0.369±0.058 | 0.643±0.080 | 0.621±0.094 | 0.686±0.081 | **1.000**±0.000 |
| | AUPR | 0.417±0.017 | 0.417±0.016 | 0.417±0.017 | 0.548±0.028 | 0.661±0.127 | 0.647±0.111 | **1.000**±0.000 | 0.455±0.054 | 0.454±0.0053 | 0.455±0.054 | 0.685±0.061 | 0.633±0.089 | 0.704±0.067 | **1.000**±0.000 |
| | FPR95 | 0.946±0.014 | 0.947±0.014 | 0.946±0.014 | 0.874±0.041 | 0.797±0.174 | 0.801±0.102 | **0.000**±0.000 | 0.996±0.003 | 0.997±0.003 | 0.996±0.003 | 0.938±0.060 | 0.919±0.046 | 0.899±0.085 | **0.000**±0.000 |

