# OpenReview forum: "Can Molecular Foundation Models Know What They Don't Know? A Simple Remedy with Preference Optimization"
_ICLR.cc/2026/Conference — Submitted to ICLR 2026_

### Official Review · Reviewer_usRx · 2025-10-31

**Soundness:** 3
**Presentation:** 3
**Contribution:** 2
**Rating:** 4
**Confidence:** 4

**Summary:**

This paper proposes Mole-PAIR, a plug-and-play post-training framework that enhances OOD detection for molecular foundation models. The key idea is to formulate OOD detection as a preference ranking task, optimizing a pairwise objective that aligns directly with AUROC. The method trains only a small scoring head on top of frozen pretrained encoders and can be applied across different models and datasets. Experiments show consistent improvements under scaffold, size, and assay distribution shifts.

**Strengths:**

1. Clear identification of a practical and important reliability issue in molecular modeling.

2. Method is architecture-agnostic, lightweight, and label-free, making it broadly usable.

3. The pairwise objective aligns well with AUROC, supported by theoretical justification.

**Weaknesses:**

1. Baseline coverage seems insufficient.
While the paper compares against several standard post-hoc OOD detection methods (e.g., MSP, ODIN, Energy, Mahalanobis, LOF, KNN), it does not include more recent and domain-specific OOD methods for molecular or graph data. This makes it difficult to determine whether the improvements stem from the proposed optimization strategy or simply from comparing against weaker baselines.

2. Lack of comparison with uncertainty-estimation models.
Deep ensembles, Monte Carlo dropout [1], SWAG [2], or conformal prediction–based uncertainty methods are widely used in high-stakes molecular prediction settings. Since the paper frames its contribution as improving reliability and mitigating hallucination, the absence of comparisons with these strong uncertainty baselines weakens the empirical claims.

[1] Representing Model Uncertainty in Deep Learning. ICML 2016.

[2] A simple baseline for bayesian uncertainty in deep learning. NeurIPS 2019.

3. Dependence on encoder embedding quality not fully discussed.
Since the encoder is frozen, the proposed method implicitly assumes that ID/OOD are separable in the pretrained embedding space. If this separability is weak, the ranking head may not compensate. The paper does not investigate encoder fine-tuning or analyze how the proposed approach performs when embedding quality varies.

4. Limited statistical and computational reporting.
The paper reports performance metrics but does not provide statistical significance across multiple seeds nor comparisons of computational cost relative to competing methods. Given the claim of cost-effectiveness, runtime and resource analysis would strengthen the empirical case.

**Questions:**

1. Have the authors evaluated whether partial or full fine-tuning of the encoder affects Mole-PAIR’s performance? If the encoder representations are not ID/OOD separable, can Mole-PAIR still provide improvements?

2. For GOOD-ZINC, what is the rationale behind median threshold binarization, and how sensitive are the results to threshold choice? Would a regression-aware scoring or continuous uncertainty metric result in different conclusions?

---

> ### Author Response · Authors · 2025-11-22
> **Response to Reviewer usRx (W1)**
>
> We thank the reviewer for acknowledging that our method is architecture-agnostic, lightweight, and theoretically well-grounded. Below we answer the weaknesses (W) and questions (Q).
>
> >W1: "Baseline coverage seems insufficient. While the paper compares against several standard post-hoc OOD detection methods (e.g., MSP, ODIN, Energy, Mahalanobis, LOF, KNN), it does not include more recent and domain-specific OOD methods for molecular or graph data. This makes it difficult to determine whether the improvements stem from the proposed optimization strategy or simply from comparing against weaker baselines."
>
> We appreciate this suggestion and agree that including domain-specific OOD methods will further strengthen our empirical evaluation. Our original comparison focuses on standard post-hoc OOD methods (e.g., MSP, ODIN, Energy, Mahalanobis, LOF, KNN) since these methods are similar to ours in the sense that they are based on the frozen encoder (architecture-agnostic) and only train a scoring head (lightweight). In contrast, the domain-specific OOD methods for molecular and graph data typically adopt different architectures or require complex generative training. That said, we supplement comparison with three recently proposed domain-specific OOD methods [1-3] and the results below show the AUROC:
>
> | Dataset/Model | PGR-MOOD [1] | GOOD-D [2] | GraphDE [3] | Mole-PAIR |
> | :--- | :--- | :--- | :--- | :--- |
> | DrugOOD-EC50-Scaffold | 0.875 +-0.013 | 0.825+-0.013 | 0.686+-0.010 | 0.970+-0.000 |
> | DrugOOD-EC50-Size | 0.977+-0.015 | 0.925+-0.013 | 0.796+-0.012 | 1.000+-0.000|
> | DrugOOD-IC50-Scaffold | 0.916+-0.013 | 0.854+-0.012 | 0.692+-0.011 |0.983+-0.000 |
> | DrugOOD-IC50-Size| 0.938+-0.015 | 0.916+-0.011 | 0.787+-0.010 |0.999+-0.000 |
>
> [1] Optimizing OOD Detection in Molecular Graphs: A Novel Approach with Diffusion Models, KDD 2024
> [2] Good-d: On unsupervised graph out-of-distribution detection, WSDM 2023
> [3] Graphde: A generative framework for debiased learning and out-of-distribution detection on graphs, NeurIPS 2022
>
> The results show that Mole-PAIR consistently outperforms these dedicated molecular or graph OOD methods, surpassing the recent SOTA method PGR-MOOD [1] by significant margins across different splits. Unlike these domain-specific models that require complex generative training with specific architectures, Mole-PAIR achieves superior performance using a simple ranking objective that only trains a lightweight scoring head.

---

> ### Author Response · Authors · 2025-11-22
> **Response to Reviewer usRx (W2)**
>
> >W2: "Lack of comparison with uncertainty-estimation models. Deep ensembles, Monte Carlo dropout, SWAG, or conformal prediction–based uncertainty methods are widely used in high-stakes molecular prediction settings. Since the paper frames its contribution as improving reliability and mitigating hallucination, the absence of comparisons with these strong uncertainty baselines weakens the empirical claims."
>
> Thank you for suggesting the comparison with these related models that can improve our evaluation. Here we add two strong uncertainty-based baselines, Monte Carlo Dropout [4] and Conformal Prediction [5], which are compatible with our setting (where we only train a scoring head on top of the foundation encoder). The AUROC results are shown as below:
>
> | Method | DrugOOD-EC50-Scaffold | Good-HIV-Scaffold |
> | :--- | :--- | :--- |
> | MSP (Baseline) | 0.677+-0.093 | 0.408+-0.054 |
> | MC Dropout | 0.728+-0.015 | 0.538+-0.055 |
> | Conformal Prediction | 0.803+-0.007 | 0.576+-0.074 |
> | Mole-PAIR (Ours)| 0.970+-0.000 | 0.777+-0.003 |
>
> We found that while MC Dropout and Conformal Prediction provide better uncertainty estimates than the simple MSP baseline, Mole-PAIR still significantly outperforms these models. The reason is that standard uncertainty methods rely on the intrinsic variance of the classifier, which often fails under distribution shifts where the model produces high-confidence false predictions (chemical hallucination). Mole-PAIR overcomes this by actively learning a discriminatory scoring function via preference optimization, allowing it to identify OOD samples based on learned relative margins rather than unreliable absolute confidence scores

---

> ### Author Response · Authors · 2025-11-22
> **Response to Reviewer usRx (W3)**
>
> >W3: "Dependence on encoder embedding quality not fully discussed. Since the encoder is frozen, the proposed method implicitly assumes that ID/OOD are separable in the pretrained embedding space. If this separability is weak, the ranking head may not compensate. The paper does not investigate encoder fine-tuning or analyze how the proposed approach performs when embedding quality varies."
>
> We appreciate this insightful comment, but we'd like to clarify that Mole-PAIR does not require the ID/OOD embeddings to be already well separated because it learns a scoring head (i.e., an MLP) that aims to maximize the separation between ID/OOD in the score space. **Given the universal capacity of MLP, even when the embeddings are not separable in the embedding space, the scoring head can be trained to map the embeddings into a well-separated score space**. Our theoretical analysis shows that the pairwise gradient emphasizes hard pairs that are near the boundary, which actively improves separability between ID/OOD in the score space even when the embeddings are far from ideal.
>
> Empirically, our experiments already involve encoders of different quality and inductive bias: MiniMol is a 2D lighter-weight encoder while Uni-Mol is a 3D more expressive encoder. In some cases, e.g., DrugOOD-IC50-Size, OOD detection performance of baselines using frozen encoders is close to random guessing, yet Mole-PAIR still achieves substantial performance gains. This indicates that the scoring head can indeed compensate for imperfect separability in the original embedding space.

---

> ### Author Response · Authors · 2025-11-22
> **Response to Reviewer usRx (W4)**
>
> >W4: "Limited statistical and computational reporting. The paper reports performance metrics but does not provide statistical significance across multiple seeds nor comparisons of computational cost relative to competing methods. Given the claim of cost-effectiveness, runtime and resource analysis would strengthen the empirical case."
>
> Thanks for carefully checking the robustness of our experimental results. In fact, **our evaluation does use multiple seeds and investigate the statistical significance**. In our original paper, due to the page limit, we defer the complete results including all performance metrics with statistical significance to **Appendix D, Table 11**. We strictly follow standard evaluation protocol: we fix the random seed for data split as 42 to ensure consistency across all experiments and perform 10 independent training runs each of which uses different random seed for weight initialization. All improvements by our model reported in the paper are statistically signifciant under $p < 0.05$ by Wilcoxon signed-rank test.
>
> To further validate the cost-effectiveness of our method, we compare the training time to convergence and GPU memory cost of Mole-PAIR against the baselines. All experiments were conducted on a single NVIDIA A100 GPU and the results are summarized below:
>
> | Method | MSP | ODIN | Energy | Mahalanobis |LOF |KNN | Mole-PAIR |
> | :--- | :--- | :--- | :--- | :--- |:--- |:--- |:--- |
> | Training Time(s) | 108.09 | 108.35 | 107.12 | 108.31 | 108.43 | 117.47 | 112.61 |
> | GPU memory (GB) | 1.52 | 1.53 | 1.52 | 1.52 | 1.52 | 1.52 | 1.6 |
>
> We found that the GPU memory cost of Mole-PAIR is comparable to all the baselines. While the pairwise loss computation results in a slight increase in training time compared to the simplest baseline MSP, this additional time cost is negligible in practice.

---

> ### Author Response · Authors · 2025-11-22
> **Response to Reviewer usRx (Q1)**
>
> >Q1: "Have the authors evaluated whether partial or full fine-tuning of the encoder affects Mole-PAIR’s performance? If the encoder representations are not ID/OOD separable, can Mole-PAIR still provide improvements?"
>
> Thanks for suggesting this comparison that can further improve our evaluation. We add comparison with partial fine-tuning (updating the last 2 transformer layers while freezing the rest) and full fine-tuning for Uni-Mol. The AUROC results are reported below:
>
> | Method | Partial fine-tuning (Uni-Mol) | Full fine-tuning (Uni-Mol)| Frozen (Uni-Mol) |
> | :--- | :--- | :--- |:--- |
> | DrugOOD-EC50-Scaffold | 0.943 | 0.833 |0.965 |
> | DrugOOD-EC50-Assay | 0.623 | 0.583 | 0.650 |
>
> We found that either partial or full fine-tuning does not bring up clear performance improvements. Our approach based on frozen encoder offers a lightweight and plug-and-play remedy for OOD detection. In contrast, full fine-tuning is computationally inefficient and introduces representation bias into the encoder that may impact the adaptability of the foundation model to other downstream tasks. The implementation details for this comparison is supplemented in Appendix C in the revised paper.
>
> For the ID/OOD separability, Mole-PAIR does not require separable embeddings of ID/OOD. We explain this in **the answer to W3** in more details.

---

> ### Author Response · Authors · 2025-11-22
> **Response to Reviewer usRx (Q2)**
>
> >"Q2: For GOOD-ZINC, what is the rationale behind median threshold binarization, and how sensitive are the results to threshold choice? Would a regression-aware scoring or continuous uncertainty metric result in different conclusions?"
>
> Binarization is used solely to adapt supervised baselines which require classification labels (including MSP, ODIN, Energy, Mahalanobis) to the GOOD-ZINC regression dataset. Mole-PAIR does not utilize task labels; it only resorts to molecular representations to compute the OOD affinity score. Thus, Mole-PAIR is independent of the binarization scheme as well as the regression label threshold. For baselines, using a median threshold is a standard practice to create balanced classes. Consequently, changing the threshold does not alter the definition of ID/OOD splits, nor does it affect the validity of Mole-PAIR’s consistent superiority over baselines.

---

> ### Comment · Reviewer_usRx · 2025-11-27
>
> Thank you for the detailed response, as well as the additional experiments and analyses. I appreciate the effort you put into addressing the earlier concerns. Most of my questions and issues have been clarified, and the new results further strengthen the contribution of the work.
>
> I encourage the authors to elegantly integrate the key findings and extended key results from the appendix directly into the main paper. Incorporating these points will help better highlight the contributions and improve the overall clarity and completeness of the manuscript.
> Based on the improvements and the clarifications provided, I have updated my score to 6 to support this work.

---

### Official Review · Reviewer_LJLj · 2025-11-01

**Soundness:** 3
**Presentation:** 3
**Contribution:** 3
**Rating:** 6
**Confidence:** 4

**Summary:**

This work aims to develop a lightweight, plug-and-play method to enhance the OOD detection capability of existing molecular foundation models without retraining the pretrained model. The authors propose Mole-PAIR, a preference optimization-based framework that reframes OOD detection as a pairwise ranking problem. Extensive experiments are conducted on multiple datasets. Results show that Mole-PAIR significantly outperforms existing OOD detection baselines.

**Strengths:**

1. This paper introduces a preference learning perspective for OOD detection in molecules, emphasizing ranking consistency between ID and OOD samples rather than per-sample confidence.
2. Mole-PAIR is model-agnostic, lightweight, and easy to deploy as a post-training module on any pre-trained molecular foundation model.
3. This paper demonstrates improvements across multiple datasets, distribution shifts, and backbone models.

**Weaknesses:**

1. Although the method does not use OOD labels during training, it still relies on pre-defined ID/OOD splits for supervision. This may limit its applicability in fully unsupervised or real-world deployment scenarios.
2. The performance is sensitive to hyperparameters β and λ, with no adaptive mechanism provided. This may require extensive tuning in practice.
3. The theoretical guarantees (e.g., convergence to Bayes-optimal ranking) assume sufficient data and model capacity. There is no analysis of performance under limited data or model misspecification, which are common in practice.
4. Are the experiments mostly conducted on binary classification tasks? I did not find evaluations of multitasking, regression, or multimodal scenarios, which are common in real-world drug discovery.

**Questions:**

Refer to Weakness

---

> ### Author Response · Authors · 2025-11-22
> **Response to Reviewer LJLj (W1)**
>
> We thank the reviewer for acknowledging the model-agnostic nature of our approach and its conceptual simplicity and strong empirical performance. Below we resolve the weaknesses (W) and questions (Q).
>
> >W1: "Although the method does not use OOD labels during training, it still relies on pre-defined ID/OOD splits for supervision. This may limit its applicability in fully unsupervised or real-world deployment scenarios."
>
> We thank the reviewer for raising this point, which was also asked by Reviewer o8ci. We respectfully clarify that in our problem setting, **the testing OOD distribution is different from the training OOD distribution** and **our method does not require access to any data from testing OOD distribution**. Furthermore, the training OOD data (which we call "Auxiliary OOD") are sampled from generic background molecules which are abundant and easy to obtain in real-world drug discovery. Our experiments are designed to evaluate the model's generalization capability, so we strictly follow the benchmarking protocols used in DrugOOD and GOOD where training OOD and testing OOD data come from different domains. The results demonstrate that our model trained with "Auxiliary OOD" data effectively generalizes to detect unseen OOD types, rather than memorizing specific splits.

---

> ### Author Response · Authors · 2025-11-22
> **Response to Reviewer LJLj (W2)**
>
> >W2: "The performance is sensitive to hyperparameters $\beta$ and $\lambda$, with no adaptive mechanism provided. This may require extensive tuning in practice."
>
> We respectfully conjecture that the perceived sensitivity is largely **a result of the visualization scale in the figures** rather than actual performance instability. In Figure 3 and 4, we utilize a logarithmic X-axis spanning three orders of magnitude alongside a highly zoomed-in Y-axis to visually amplify the variation of Y w.r.t. change of X. Quantitatively, the performance is remarkably stable w.r.t. change of $\beta$. In Figure 3, as $\beta$ varies from 0.01 to 10, the AUROC fluctuation is less than 0.02 on Scaffold splits and nearly negligible on Size splits. Regarding $\lambda$ in Figure 4, the performance drop at excessively high values ($\lambda > 0$) is theoretically expected as over-regularization suppresses the ranking scores. However, Figure 4 clearly shows that the model performance remains stably optimal within a wide range of $\lambda$ from 0.01 to 0.1.
>
> More importantly, we do not perform extensive hyper-parameter tuning for each task in our experiments. The main results across 5 datasets, 3 shift types, and 2 backbones are achieved using a default configuration ($\beta = 0.1$, $\lambda = 0.01$). Even with this single setting, the model consistently achieves SOTA performance across all cases, suggesting that **extensive hyper-parameter tuning is not required in practice**.

---

> ### Author Response · Authors · 2025-11-22
> **Response to Reviewer LJLj (W3)**
>
> >W3: "The theoretical guarantees (e.g., convergence to Bayes-optimal ranking) assume sufficient data and model capacity. There is no analysis of performance under limited data or model misspecification, which are common in practice."
>
> We agree that the theoretical guarantees are contingent upon the assumptions about sufficient data and model capability, which are in fact the common assumptions adopted in the literature. That said, the training of Mole-PAIR is data-efficient, as supported both theoretically and empirically. Theoretically, the gradient update of Mole-PAIR naturally prioritizes hard examples that are near the boundary (Proposition 4.1) and assigns larger weights to informative ID-OOD pairs, leading to effective learning even with scarce data (as validated by visualization in Appendix D.2). Empirically, our experiments do verify the model's efficacy in the low-data regime: for DrugOOD, we use only 2000 ID and 2000 OOD training samples per split. Despite this, Mole-PAIR consistently yields significant improvements across all cases, suggesting its robustness to limited training data.
>
> Regarding model misspecification, our method relies on post-hoc score calibration rather than reshaping the encoder’s representation space. By freezing the foundation model and only training a lightweight scoring head, Mole-PAIR is agnostic to the architectures of representation encoders. In our experiments, we apply Mole-PAIR to two encoders MiniMol (2D) and Uni-Mol (3D) which differ substantially in their architectures, and Mole-PAIR achieves superior results using both backbones. These results altogether show that Mole-PAIR is resilient to data scarcity and different encoder backbones, even though our theoretical analysis follows the standard assumptions.

---

> ### Author Response · Authors · 2025-11-22
> **Response to Reviewer LJLj (W4)**
>
> >W4: "Are the experiments mostly conducted on binary classification tasks? I did not find evaluations of multitasking, regression, or multimodal scenarios, which are common in real-world drug discovery."
>
> We'd like to clarify the potential misunderstanding into our method. **Mole-PAIR is a task-agnostic approach that does not utilize any labels from downstream tasks**. The model uses molecular representations derived from raw molecular inputs (e.g., SMILES strings or 3D conformations) to compute OOD scores and is learned from the ID/OOD ranking. Therefore, the model is applicable to binary and non-binary classification tasks, regression tasks, multitask settings, and any data modality.
>
> In fact, our experiments already include non-binary and non-single-task scenarios, as detailed in Appendix C.1:
> - **Multitasking (GOOD-PCBA)**: GOOD-PCBA is a large-scale dataset containing 128 bioassays. This is a standard multi-task prediction benchmark.
> - **Regression (GOOD-ZINC)**: GOOD-ZINC is a molecular property regression dataset which aims at predicting constrained solubility.
>
> Extending Mole-PAIR to the multimodal setting is straightforward, given its task-agnostic nature. For example, one can apply Mole-PAIR to a multimodal molecular foundation model (e.g., Sequence–Structure joint encoders) and the scoring head simply aggregates the encoder output of different modalities. We leave exploration in this orthogonal direction as future work.

---

### Official Review · Reviewer_o8ci · 2025-11-02

**Soundness:** 3
**Presentation:** 3
**Contribution:** 3
**Rating:** 4
**Confidence:** 4

**Summary:**

The paper proposes a method for ranking in-distribution samples ahead of out-of-distribution samples. An MLP is finetuned on a frozen encoder such as
MiniMol and a preference learning objective is uesd to maximize the AUROC. Experiments on multiple datasets in different settings demonstrate the
performance of the proposed method, showing high performance gains. Some convergence guarantees are also provided for the proposed method.

**Strengths:**

- The proposed method is model agnostic allowing it to be applied to any pretrained model of interest. The cost of training itself is low given that a
  small MLP is trained with the foundation model frozen. This allows for an easy remedy to fixing the OOD issues in foundation models.

- The theoretical guarantees provided makes the model sound.

- Performance wise, the proposed model shows good empirical results over a range of datasets across multiple data shift regimes.

**Weaknesses:**

- In real world scenarios, the assumption of an OOD dataset to use in the loss function may not be realistic. In most cases, one does not have access
  to the OOD samples that one wishes to generalize on. At any rate, even if such a split exists, in practice it may be continually shifting.
  Additionally, there are many situations where crafting such an OOD datasets may not even be feasible. Hence it is unclear as to how this method can
  be adapted to generalize to unseen OOD samples.

- The proposed loss term requires pair-wise comparisons of ID and OOD samples. Practically, this can be a huge space to explore and an analysis of
  performance compared to how much this pair-wise space is explored can be useful in gauging how cost effective the method actually is as claimed in
  the paper.

- In the current framework, the encoder is frozen and no comparison is provided on how the proposed method compares on a simple finetuning or even
  contrastive approach which seems a natural choices here given that the ID and OOD splits are assumed given.

**Questions:**

Together with some of the weaknesses discussed,

- Currently, the proposed method assumes access to ID and OOD data splits in the post training phase. How realistic is this in the drug discovery
  pipeline? What happens when we do not have access to an OOD dat asplit? Suppose we have access only to the ID split then is there a way to use the
  pairwise approach via some other pairwise scheme that allows for OOD generalization? And does Mole-Pair only generalize on given OOD data?

---

> ### Author Response · Authors · 2025-11-22
> **Response to Reviewer o8ci (W1&Q1)**
>
> We thank the reviewer for recognizing the cost-effective plug-and-play characteristics of our approach as well as its theoretical motivation and strong empirical performance. We also thank the reviewer for the critical questions regarding the data assumption and practicality. Below we answer the weaknesses (W) and questions (Q) where we merge similar points as a single response.
>
> >W1: "In real world scenarios, the assumption of an OOD dataset to use in the loss function may not be realistic. In most cases, one does not have access to the OOD samples that one wishes to generalize on. At any rate, even if such a split exists, in practice it may be continually shifting. Additionally, there are many situations where crafting such an OOD datasets may not even be feasible. Hence it is unclear as to how this method can be adapted to generalize to unseen OOD samples."
>
> > Q1: "Currently, the proposed method assumes access to ID and OOD data splits in the post training phase. How realistic is this in the drug discovery pipeline? What happens when we do not have access to an OOD datasplit? Suppose we have access only to the ID split then is there a way to use the pairwise approach via some other pairwise scheme that allows for OOD generalization? And does Mole-Pair only generalize on given OOD data?"
>
> Thanks for bringing up this important question which allows us to clarify our problem setting. The reviewer raises a critical point: assuming access to the OOD data for training is unrealistic in real-world scenarios, with which we fully agree. However, we would like to clarify that **the testing OOD distribution is different from the training OOD distribution** and **our method does not require access to any data from testing OOD distribution**.
>
> In our setup, $D_{out}^{train}$ represents "Auxiliary OOD", which means that they are a proxy for the general chemical background rather than the specific OOD types we aim to detect during testing. In drug discovery scenarios, while specific bioactivity data (i.e., ID data) is expensive and scarce, generic molecular data is abundant. For the latter, we have access to millions of molecules in public databases (e.g., ZINC, PubChem, ChEMBL, etc.) and sampling from them as chemically diverse background molecules to define **''what is not ID''** is both realistic and standard. **Our method leverages this abundant generic molecule data to compose the "Auxiliary OOD" (i.e., the training OOD data)** rather than relying on peeking at any data from the testing OOD distributions.
>
> To further resolve the lingering concern on Mole-PAIR's generalization ability, especially whether it can extrapolate to detect OOD data from entirely different domains, we supplement new experiments where the model needs to generalize from one dataset of molecules (used for training and validation) to another new dataset (for testing). Specifically, we train and validate Mole-PAIR on DrugOOD-EC50-Scaffold, and test it on DrugOOD-IC50-Size. In addition, we also evaluate the reverse setting by training on the Size split and testing on the Scaffold split to ensure bidirectional robustness. The results are shown below:
>
> | Training Set | Test Set | AUROC | AUPR | FPR95 |
> | :--- | :--- | :--- | :--- | :--- |
> | DrugOOD-EC50-Scaffold | DrugOOD-IC50-Size | 0.968+-0.001 | 0.973+-0.001 | 0.196+-0.003 |
> | DrugOOD-EC50-Size | DrugOOD-EC50-Size | 1.000+-0.000 | 1.000+-0.000 | 0.000+-0.000 |
> | DrugOOD-IC50-Scaffold | DrugOOD-EC50-Size | 0.984+-0.001 | 0.986+-0.001 | 0.069+-0.003 |
> | DrugOOD-IC50-Size | DrugOOD-IC50-Size | 0.999+-0.000 | 0.999+-0.001 | 0.004+-0.001 |
>
> This shows that even when the testing OOD data are generated from an entirely different distribution than the training OOD data, the model can still generalize well.

---

> ### Author Response · Authors · 2025-11-22
> **Response to Reviewer o8ci (W2)**
>
> >W2: "The proposed loss term requires pair-wise comparisons of ID and OOD samples. Practically, this can be a huge space to explore and an analysis of performance compared to how much this pair-wise space is explored can be useful in gauging how cost effective the method actually is as claimed in the paper."
>
> We understand the reviewer's concern regarding the potential computational cost of exploiting the full pairwise space. We respectfully clarify that **our proposed loss function does not require exhaustive exploitation of the global $N_{ID} \times N_{OOD}$ pairwise space**. Instead, we employ a stochastic mini-batch pairing strategy that is computationally efficient and scalable: for each training iteration, we randomly sample a mini-batch of size $B$ from the ID dataset and a mini-batch of size $B$ from the OOD dataset, and then form pairs within this mini-batch which leads to $B \times B$ pairs. This yields $O(B^2)$ complexity per iteration, which is independent of the total dataset size. In practice, we found using a small batch size (e.g., 32) works consistently well across difference datasets, which enables efficiency for training. Furthermore, our empirical results (Appendix D) validate that this stochastic sampling effectively approximates the gradients of the global pairwise distribution, enabling the model to learn robust decision boundaries without needing to enumerate the full $N_{ID} \times N_{OOD}$ space.

---

> ### Author Response · Authors · 2025-11-22
> **Response to Reviewer o8ci (W3)**
>
> >W3: "In the current framework, the encoder is frozen and no comparison is provided on how the proposed method compares on a simple finetuning or even contrastive approach which seems a natural choices here given that the ID and OOD splits are assumed given."
>
> We appreciate the reviewer's suggestion to compare with finetuning and contrastive methods.
>
> **Comparison with finetuning.** We'd like to clarify that our experiments already include the "simple finetuning" baselines. As described in Section 5, all supervised methods such as MSP, Energy, and ODIN are implemented by attaching an MLP classifier head to the frozen encoder and training this head with standard CrossEntropy/BCE/MSE objectives. This corresponds exactly to the "finetuning on top of frozen features" strategy that is widely used in molecular property prediction. That said, to further validate the effectiveness and superiority of our proposed approach, we supplement comparison with partial fine-tuning foundation models (Uni-Mol) and full fine-tuning foundation models (Uni-Mol). The AUROC results are reported as below (implementation details are in Appendix C in the revised paper.):
>
> | Method | Partial fine-tuning (Uni-Mol) | Full fine-tuning (Uni-Mol)| Frozen (Uni-Mol) |
> | :--- | :--- | :--- |:--- |
> | DrugOOD-EC50-Scaffold | 0.943 | 0.833 |0.965 |
> | DrugOOD-EC50-Assay | 0.623 | 0.583 | 0.650 |
>
> This reveals that neither partial nor full parameter tuning significantly enhances results, but largely increasing the training cost and time. Consequently, our proposed method utilizes a fixed encoder, which offers a lightweight and versatile remedy for OOD tasks. Furthermore, by freezing the foundation model, we prevent the introduction of bias into the representation space, preserving the foundation model's adaptability for future tasks.
>
> **Comparison with contrastive approach.** We also acknowledge the connection between our objective and contrastive learning. Mole-PAIR can be viewed as a pairwise contrastive method, but with a crucial difference in the optimization target. Standard contrastive methods operate in the representation space, encouraging embeddings of similar samples to be close and dissimilar ones to be apart, without directly targeting the OOD detection metric or the score distributions themselves. In contrast, **Mole-PAIR represents a form of contrastive learning directly in the score space:** it optimizes a pairwise objective that explicitly enforces a margin between ID and OOD scores and is theoretically aligned with the AUROC criterion. However, extending standard contrastive approaches to our case is not straightforward, which makes it hard for reasonable comparison in our setup.

---

### Official Review · Reviewer_LnWv · 2025-11-03

**Soundness:** 2
**Presentation:** 3
**Contribution:** 2
**Rating:** 4
**Confidence:** 2

**Summary:**

This work introduces Mole-PAIR, a post-training framework to improve the out-of-distribution (OOD) detection ability of molecular foundation models. The key idea is to reformulate OOD detection as a preference optimization problem, where a pairwise ranking objective is optimized instead of traditional pointwise regression or confidence estimation. The method trains a lightweight MLP detector on top of frozen pretrained encoders such as Uni-Mol and MiniMol. Theoretical analyses show that the proposed loss aligns with optimizing AUROC, and extensive experiments on DrugOOD and GOOD benchmarks demonstrate large improvements in OOD detection under multiple distribution shifts.

**Strengths:**

The paper addresses an important and practical challenge in molecular AI, which is how to make molecular foundation models more reliable under distribution shifts. The formulation is conceptually clean, and the idea of aligning the training objective directly with AUROC is theoretically well motivated. The method is simple, efficient, and compatible with existing pretrained molecular encoders without requiring retraining. The theoretical analysis is complete and clearly written. The experiments are broad and consistent across datasets, showing that the approach can yield strong performance improvements under various types of shifts.

**Weaknesses:**

The proposed framework mainly applies the pairwise ranking principle, long established in AUROC optimization and preference learning, to the molecular OOD detection setting. Theoretical analysis provides clarity but does not introduce new insights beyond known ranking formulations.

**Questions:**

How sensitive is Mole-PAIR to the availability of OOD samples during post-training? Does performance degrade when OOD examples are scarce or unbalanced relative to ID data?

---

> ### Author Response · Authors · 2025-11-22
> **Response to Reviewer LnWv (W1)**
>
> Thank you for acknowledging the practical significance of the problem we address and the theoretical soundness of our proposed objective. We also appreciate your recognition of our method's simplicity and effectiveness under distribution shifts. Below we answer the weaknesses (W) and questions (Q).
>
> > W1: "The proposed framework mainly applies the pairwise ranking principle, long established in AUROC optimization and preference learning, to the molecular OOD detection setting. Theoretical analysis provides clarity but does not introduce new insights beyond known ranking formulations."
>
> We respectfully disagree with this interpretation of our contributions. While pairwise ranking has been well established in other domains such as information retrieval, our unique contribution represents a paradigm shift for molecular OOD detection, i.e., a new emergent problem in the foundation model era. We elaborate our novelty and significance from three aspects:
>
> - **Paradigm Shift: from pointwise to pairwise OOD objectives.** Existing molecular OOD methods predominantly rely on pointwise confidence estimation, which leads to the misalignment between the training objective (e.g., BCE/MSE) and the evaluation metric (AUROC). Our preliminary results in Figure 1 serve as empirical evidence for this observation. Our contribution lies in that **we are the first to re-formulate the OOD detection problem in molecular foundation models as a preference optimization task**. By introducing the pairwise learning paradigm to this new domain, we resolve the long-standing Objective-Metric Misalignment where models trained with BCE/MSE fail to directly maximize AUROC.
>
> - **Theoretical Insight.** Our analysis elucidates the mechanism of our approach in this new under-explored problem. We derive the gradient dynamics of Mole-PAIR and prove that it naturally prioritizes "hard" and "boundary" pairs (Proposition 4.1). This shows that unlike standard ranking which treats pairs equally, Mole-PAIR can adaptively focus the optimization on "hard" examples that facilitate learning good OOD detector. **This provides theoretical grounds supporting that pairwise learning is provably effective for OOD detection** (which is further empirically validated by the gradient weight analysis in Appendix D.2).
>
> - **Justification of Significance.** We believe that high-impact research often lies in identifying a simple method with theoretical rigor for an emerging, urgent problem. Chemical hallucination in molecular foundation models is a critical bottleneck for drug discovery, yet it lacks a specialized solution that is both theoretically sound and practically effective. **We are the first to bridge the gap between classic preference learning and molecular OOD detection**. We do not simply apply the theory and instead we extend it by connecting the ranking loss with AUROC maximization in the new context. This leads to a principled, model-agnostic and lightweight remedy for a new problem, avoiding the complexity of generative or ensemble-based methods. Similarly, many impactful advancements in machine learning, such as RLHF [1], which aligns LLMs using the classical Bradley-Terry model, and Diffusion Models [2,3], which apply thermodynamics to generative modeling, stem from identifying the connection between classic methods and the pressing challenges at the moment. We believe Mole-PAIR represents a similarly significant contribution in the domain of AI reliability for science.
>
> [1] Training language models to follow instructions with human feedback, NeurIPS 2022
>
> [2] Deep unsupervised learning using nonequilibrium thermodynamics, PMLR 2015
>
> [3] Denoising Diffusion Probabilistic Models, NeurIPS 2020

---

> ### Author Response · Authors · 2025-11-22
> **Response to Reviewer LnWv (Q1)**
>
> > Q1: "How sensitive is Mole-PAIR to the availability of OOD samples during post-training? Does performance degrade when OOD examples are scarce or unbalanced relative to ID data?"
>
> Thanks for raising this point which allows us to clarify the potential misunderstanding on our model. Mole-PAIR does not require a large, diverse set of OOD examples. Since its learning objective aims at preserving the relative ranking between ID and OOD samples, a few OOD examples can provide effective guidance during post-training. The gradient weighting property (Proposition 4.1) enables the training process to naturally focus on informative pairs, ensuring robustness even when OOD data is limited or imbalanced.
>
> To validate this claim, we supplement additional experiments manipulating the ID:OOD ratio in the training data. We reduce the number of training OOD samples while fixing the number of ID samples as 2,000 (validation and test splits remained unchanged). We vary the training ID:OOD ratio from 1:1 to 1:0.01, simulating increasingly scarce OOD availability. The AUROC results for Mole-PAIR are shown below:
>
> | ID:OOD Ratio | 1:1 | 1:0.1 | 1:0.02 | 1:0.01 |
> | :--- | :--- | :--- | :--- | :--- |
> | DrugOOD-EC50-Scaffold | 0.970+-0.000 | 0.924+-0.001 | 0.898+-0.003 | 0.849+-0.005 |
> | DrugOOD-EC50-Size | 1.000+-0.000 | 0.993+-0.001 | 0.991+-0.001 | 0.974+-0.020 |
> | DrugOOD-EC50-Assay | 0.711+-0.002 | 0.638+-0.005 | 0.621+-0.004 | 0.603+-0.002 |
>
> We found that the performance degradation is moderate even when training OOD samples become extremely scarce (e.g., only 20 training OOD samples for the 1:0.01 ratio). Notably, even using the ID:OOD ratio of 1:0.01, Mole-PAIR still outperforms the strongest baseline trained with the 1:1 ratio (which is reported in Table 1). These serve as strong empirical evidence that Mole-PAIR is robust to limited and imbalanced OOD availability, remaining a cost-effective post-training remedy even with very few OOD samples.

---

### Author Response · Authors · 2025-11-22
**General Response to All Reviewers**

Dear Area Chairs and Reviewers,

We'd like to thank all the reviewers for their thorough reviews, constructive comments, and nice suggestions that improve our work. We are pleased that the reviewers acknowledged the significance of our studied problem (Reviewer LnWv, Reviewer usRx), our theoretically sound and clean formulation (Reviewer LnWv, Reviewer 08ci, Reviewer usRx), the model-agnostic and lightweight approach (Reviewer LnWv, Reviewer 08ci, Reviewer LJLj, Reviewer usRx), and its solid empirical performance across various datasets (Reviewer LnWv, Reviewer 08ci, Reviewer LJLj).

The major questions raised by the reviewers lie in the **practical assumptions regarding OOD data availability and splits** (Reviewer LnWv , Reviewer 08ci , Reviewer LJLj ), **more baseline comparisons particularly regarding uncertainty estimation and fine-tuning methods** (Reviewer 08ci , Reviewer usRx), and **model robustness and sensitivity w.r.t. encoder quality** (Reviewer LJLj , Reviewer usRx ). In our individual response below, we answer these questions as well as other minor points in detail.

Furthermore, in light of the suggestions from the reviewers, we supplement a list of new experiments (see results in the individual response):

- Model performance under different levels of OOD data scarcity. (**Appendix Table 5**)
- Evaluation of generalization capabilities on unseen OOD types. (**Appendix Table 6**)
- Comparison with domain-specific OOD methods for molecular data and uncertainty-estimation models. (**Appendix Table 7 and Table 8**)
- Comparison of computation costs including running time and GPU memory for different models. (**Appendix Table 9**)
- Impact of encoder fine-tuning strategies on model performance. (**Appendix Table 10**)

We also revise the paper accordingly with these new results incorporated. We are happy to provide further clarification and additional experiments to resolve any further question or feedback during the discussion.

Thank you again for taking the time to review our paper.

---

### Author Response · Authors · 2025-11-24

Dear Reviewers,

We’d like to thank you again for your thorough review and constructive feedback. In light of the comments, we have added an extensive set of new results (see general response for a summary and individual responses for detailed results) and revised our paper accordingly.

We are sincerely looking forward to your further feedback and hoping that you can re-evaluate the ratings considering our response and revised paper.

---

### Comment · Area_Chair_b3QZ · 2025-11-26

Dear reviewers,

The authors have provide updates and clarifications during the rebuttal, and they are awaiting your follow-up comments.

Please take a moment to review the updates.

Thank you for your efforts.

---

### Meta-Review · Area_Chair_TQx9 · 2026-01-06

**Summary:**

The paper introduces Mole-PAIR, a post-training framework designed to enhance the out-of-distribution (OOD) detection capabilities of molecular foundation models by reformulating the task as a preference optimization problem. While the reviewers recognized the practical significance of improving reliability in molecular AI and appreciated the model-agnostic, lightweight nature of the proposed remedy, concerns stand out regarding the limited technical novelty of the approach and the reliance on auxiliary OOD samples. Specifically, reviewers noted that the core methodology—applying pairwise ranking principles to optimize AUROC—is a well-established concept in preference learning and AUROC optimization. Furthermore, the framework's reliance on auxiliary OOD samples during training was flagged as a potential limitation for real-world scenarios where representative OOD data may not be readily available, raising questions about the method's practical superiority over unsupervised or in-distribution-only approaches.

**Reviewer Concerns:**

Throughout the rebuttal period, the authors were highly proactive in addressing technical and empirical gaps identified by the reviewers.

**Addressed**:
The authors successfully incorporated extensive new comparisons with domain-specific molecular OOD methods (e.g., PGR-MOOD, GOOD-D) and uncertainty-estimation baselines like Monte Carlo Dropout and Conformal Prediction, demonstrating that Mole-PAIR consistently outperforms these alternatives. They also provided a detailed computational cost analysis showing the method's efficiency in terms of training time and GPU memory. Additionally, they clarified the distinction between "Auxiliary OOD" used for training and the unseen OOD distributions used for testing.

**Outstanding**:
The fundamental concern regarding incremental novelty remains outstanding; while the authors argued that their work represents a "paradigm shift" by connecting classic methods to a new domain, the underlying theoretical analysis was not seen as providing significant new insights beyond existing ranking formulations. Moreover, the theoretical justification for "auxiliary OOD" selection remains a point of skepticism, as the ability of a limited set of background molecules to generalize to arbitrary, high-dimensional chemical shifts was not fully validated beyond the provided benchmarks.

**Reviewer Scores:**

Reviewer  **usRx** (Initial Score 4) : This reviewer already increased their score after the authors addressed concerns regarding baselines and fine-tuning.

Reviewer **LJLJ** (Initial Score 6): Likely would have maintained their score, as the authors clarified the task-agnostic nature of the model and its applicability to multi-task and regression benchmarks.

Reviewer **LnWv**	(Initial Score 4) : The primary concern was a lack of technical novelty, which the rebuttal characterized as a "paradigm shift". The AC partially agree with the authors on the novelty of the work. The score is likely to be changed to 5.

Reviewer **o8ci** (Initial Score 4):	Might have moved to a 5 after seeing the new generalization experiments , but the core philosophical objection to requiring OOD training data would likely prevent a score of 6.

---

### Decision · Program_Chairs · 2026-01-26

Reject